# Individual differences in how infants change behaviours from spontaneous to instrumental
Ryo Fujihira ✉, Hama Watanabe & Gentaro Taga

Young infants can change their behaviour and learn through interactions with novel environments. This ability has been demonstrated through group-averaged analyses. However, it remains unclear whether averaged behavioural changes accurately capture the diverse changes occurring at the individual level. To address this, we measured limb movement alterations in 185 infants aged 2 to 3 months before and after their arm was tethered to an overhead mobile and analysed individual differences in addition to conventional group analyses. While the group-averaged data showed a gradual increase in arm movements, individual learning curves rarely exhibited such simple gradual increases and instead displayed more complex patterns. To disentangle the complex movement patterns, we applied time-series clustering and dynamical systems modelling to our large-scale dataset. As a result, the infants were divided into distinct clusters with significant differences in spontaneous movements before learning, rather than after. A dynamical systems model further demonstrated that only differences in spontaneous movements could explain the diversity of overall behavioural changes. These findings indicate that the varying degrees of behavioural change reflect infants' unique learning processes rather than their learning capabilities. Furthermore, learning, as a process that reduces individual difference, suggest that infants harness their unique spontaneous movements to acquire instrumental behaviours.

The ability of young infants to learn in the world has been inferred through measurements of changes in their behaviour when they encounter novel environmental events. They are sensitive to contingencies between their own movements and environmental events, making changes in their behaviours, such as limb movements with attractive mobile motion[1–4], arm movements with facial images of the mother[5], eye movements with smiling faces[6], and non-nutritive sucking with the mother's voice[7]. While the group analyses of such behavioural changes have supported the existence of learning abilities in infants as early as 2 months of age, it remains unclear whether the averaged behavioural changes accurately capture the mechanisms underlying changes at the individual level. Previous studies have demonstrated individual differences in the developmental trajectories of motor and behavioural skills over timescales of days, months or years. These include the acquisition of reaching[8–10], the development of crawling[11], sitting[12] and walking[12,13], as well as more general processes of skill learning[14]. However, inter-individual differences in learning over shorter timescales have not been clearly identified. Short-term learning is not a mere fluctuation for development, but potentially drive longer-term development through memory processes[15], and dynamic interactions between brain, body, and environment. In particular, the behaviour of young infants is characterised by varieties of spontaneous movements, which evolve rapidly over the postnatal period[16] but exhibit intra-individual consistency alongside significant inter-individual differences[17–19]. Through these movements, infants can initiate interactions with the physical world leading to acquire actions along their unique ways[20,21]. Thus, different patterns of spontaneous movements can provide different foundations for shaping actions toward environments. In the context of learning, we cannot ignore the fact that intrinsic movement repertoires may differ from infant to infant, which raises the following question: how do the patterns of intrinsic spontaneous movements affect learning.

To examine the dynamic evolution of behaviours during learning, we leveraged an experimental paradigm known as the mobile paradigm, which has widely been used for revealing learning capabilities in early infancy[1,4,22–25]. In this paradigm, an infant's limb is tied to an overhead mobile with a ribbon and their limb movements cause the mobile to move. As a result of learning contingency between their own movements and mobile movements, infants change their spontaneous movements into actions suitable for moving the mobile. We refer to the learned actions as

Graduate School of Education, The University of Tokyo, Tokyo, Japan. ✉e-mail: fujihira@p.u-tokyo.ac.jp; fujihira@nips.ac.jp

instrumental behaviours[26], as they are performed to achieve a specific outcome. Although such behaviours are probably goal-directed[4], it is uncertain whether infants have a goal when performing these actions. Thus, we adopt the term 'instrumental'. The increase in the connected limb movement is interpreted as an indication of learning, and group analyses have shown that learning is established in infants as early as 2 months of age[1,2,27,28]. Based on these results, it has become customary to use an increase of 1.5 times the baseline level of limb movement as the threshold for learning[3,29,30]. However, a more detailed analysis of pre-learning states and changes during the learning process have suggested that learning involves dynamic properties that cannot be solely captured by the reinforcement of movement. For example, some previous studies have demonstrated that different groups of infants, at varying months or days of age, exhibited distinct changes in limb movement patterns during the learning process[23,31]. Even within the same age group, the increase of movement differed between groups divided by the frequency of spontaneous movements prior to learning[32]. These studies imply that learning is not solely reflected in increased limb movement. Nonetheless, the idiographic features of behavioural changes have not been fully uncovered. Therefore, this study aims to investigate individual differences in behavioural trajectories, with the fundamental goal of identifying nomothetic principles that apply across individuals. To achieve this, we set up groups of a large number of 2- and 3-month-old infants with precise age distributions ranging 10 days and conduct a mobile task, analysing individual differences in the time evolution of their movements. Unlike previous studies that measured the number of kicks, this study attached a string to the arm and employed a motion analysis system to capture richer movement dynamics.

Our analysis comprises three sections: a conventional group-averaged analysis, a clustering analysis to explore individual differences in behavioural changes, and dynamical systems modelling to explain these differences. In the first section, we examine the changes in the amount of infants' limb movement after the mobile connection and differences between 2- and 3-month-old infants at the group level. We also examine the features of individual differences in limb movements before and after learning, as well as in the behavioural changes between them, by plotting individual data. This analysis further investigates whether age differences alone can fully account for individual differences in learning. In the next section, we apply time-series clustering to the data of the time evolution of all limb movements in order to extract the variety of infants' behavioural changes. While individual differences in behavioural changes during the mobile paradigm have been studied by closely observing the time evolution of limb movement alterations in each infant[20,26,33–35], this method is inapplicable to a large-scale dataset. Instead, we utilise clustering methods to divide the large number of infants into several clusters and analyse individual differences along with the clusters. In the last section, we explore a possible mechanism underlying the learning process with individual differences by using a dynamical systems model for the mobile paradigm[36]. In the dynamical systems approach, spontaneous movements are captured by the concept of intrinsic dynamics, from which new coordinated behaviours emerge[36–38]. Observed behavioural states are generated from a dynamical system composed of an infant's brain, body, and environment, containing the overall characteristics of each infant's movement dynamics. Therefore, the behavioural trajectory of each individual is considered to reflect the individual differences in the structurally stable attractor dynamics underlying movement generation. We

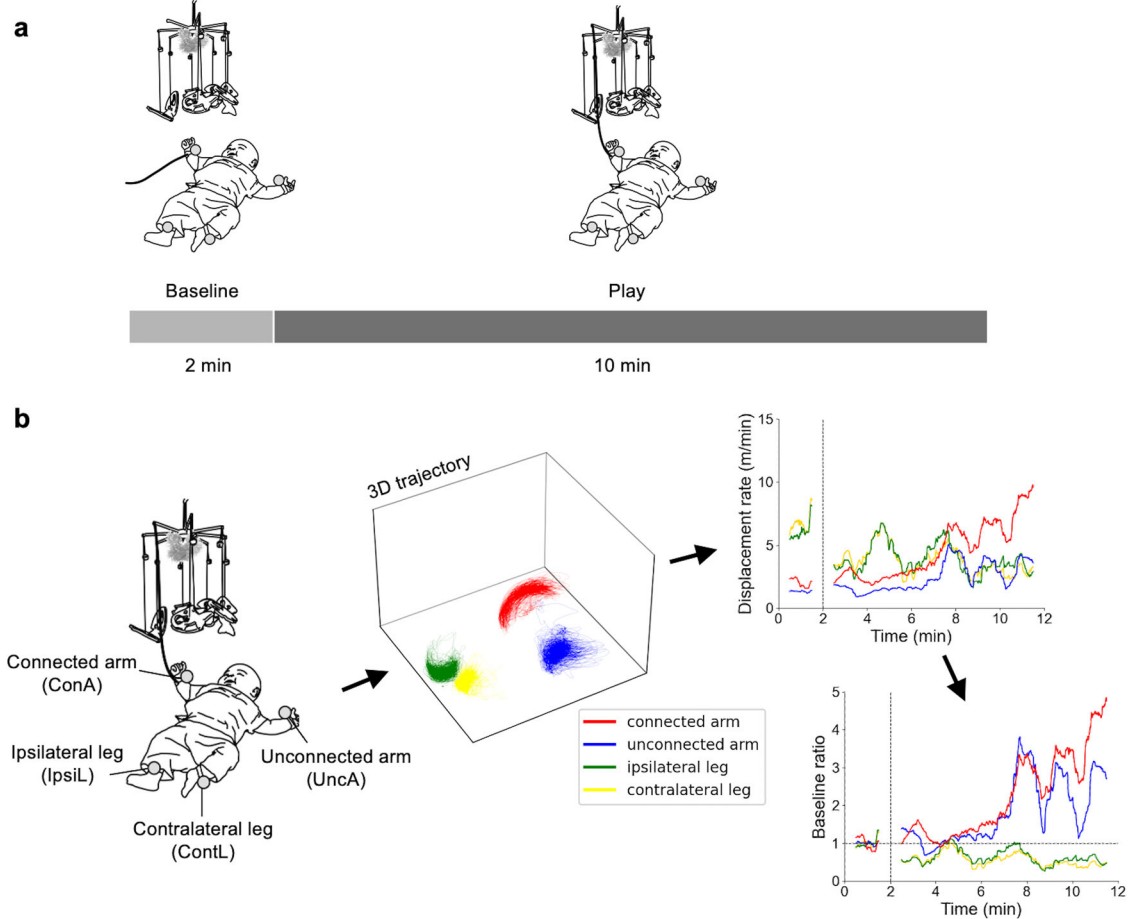

**Fig. 1 | Schema of experiment. a** Experimental procedure. **b** The procedures of data acquisition and data preparation. Displacement rate represents the amount of movement in each limb in a 1 min time window. Baseline ratio is the ratio of the displacement rate to its averaged value in the baseline phase. Thus, the average of the baseline ratio over the baseline phase is equal to 1.

leverage the concept of dynamical systems by mapping the various patterns of limb movement alterations onto differences in a parameter that determines the degree of spontaneous movements. We first simulate individual trajectories of behavioural changes over the course of learning, and then use them to reproduce the learning curves at the group level.

## Methods

### Participants

To elucidate age-related behavioural differences and thoroughly examine individual differences, we targeted two age groups: 2-month-olds (70–79 days) and 3-month-olds (101–109 days) with a total of 274 participants, which is considered a relatively large sample size for experiments of this type with young infants[39]. 185 infants completed the experimental procedures (2-month-old infants: $n = 90$, 49 girls and 51 boys, age 70–79 days, mean age 75.1 ± 2.6 days; 3-month-old infants: $n = 95$, 40 girls and 55 boys, age 101–109 days, mean age 105.3 ± 2.1 days) Additional 94 infants participated in the present study but were excluded from the sample because of crying or fussing ($n = 64$), rolling over ($n = 5$), drowsiness ($n = 2$), finger sucking ($n = 4$), hiccup ($n = 1$) or system errors ($n = 18$). All participants were full-term infants recruited via the local Basic Resident Register. Ethical approval for this study was obtained from the ethical committee of Life Science Research Ethics and Safety, the University of Tokyo, and written informed consent was obtained from the parent(s) of all the infants prior to the initiation of the experiments. No aspects of this experiment were preregistered.

### Apparatus

All measurements were conducted in the laboratory at the Graduate School of Education of the University of Tokyo. A baby mattress (120 cm × 70 cm), a non-commercial handmade mobile (length, 60 cm), and a modified clothes hanger stand (height of the bar from the baby mattress, 90 cm) were used for the experiments. The experimental setup was identical to that used in previous studies[23,28,31,32,40,41]. Each infant was positioned on their back on the baby mattress, with the mobile suspended above the infant from the hanger stand. During the baseline phase, infants moved spontaneously, observing the hand-made mobile, which consisted of eight coloured objects with 16 small bells. During the play phase, a string was attached between the mobile and the right or left wrist of the infant. Thereby, the related movement of the arm induced the movements of the coloured objects and sounds from the bells.

The infants' limb movements in the three-dimensional (3D) space were recorded using a real-time 3D motion capture system (Motion Analysis Co., Santa Rosa, California). Six CCD monochrome shuttered cameras (60 Hz; Hawk i Digital Camera) with electronically shuttered infra-red LED synchronised strobe lighting were placed around the baby mattress. Spherical reflective markers: 2 cm in a diameter weighting approximately 5 g, were attached to the wrists and ankles of the infant in order to track the position of them. Additionally, three video cameras were used to record other information, such as facial expressions, gross movements and the status of the mobile.

### Procedures

Prior to the experiment, infants were positioned on their back within the test field and were attached four reflective markers to their wrists and ankles. As the infants were likely to be alert and playful, the measurements were carried out. The first 2-min period of the measurement was a baseline phase in which infants' spontaneous movements did not cause the mobile to move, observing the motionless mobile suspended above. After the baseline phase, a string was attached between the mobile and an infant's wrist (the connected wrist, right or left, was counterbalanced between the participants). This arrangement initiated a play phase lasting 10 min. During the play phase, any movements of the connected arm, like wiggling, pulling, waving, and so on, produced a corresponding degree of movement in the overhead mobile (Fig. 1a). The measurement duration of the current study is longer than our past studies[23,28,31,32,40,41] in order to capture the full trajectory of

behaviour while infants learn the contingency with the environment. In the mobile paradigm, an extinction phase is typically introduced at the end by removing the connection between the infant's limb and the mobile. In the present study, however, we extended the play phase instead, as our primary objective was to investigate the variety of behavioural trajectories from spontaneous to instrumental. In this context, we named the phase typically referred to as the acquisition phase as the play phase. This terminology reflects a broader conceptualisation of learning—not only the acquisition of specific movements but also including a variety of intermediate behavioural changes.

### Data preparation

The time series for 3D positions ($x$, $y$, $z$) of four limbs were obtained with a sampling frequency of 60 Hz. We calculated the instantaneous 3D velocity from the positions using the central difference method:

$$V_t = \sqrt{\left(\frac{x_{t+1} - x_{t-1}}{2T}\right)^2 + \left(\frac{y_{t+1} - y_{t-1}}{2T}\right)^2 + \left(\frac{z_{t+1} - z_{t-1}}{2T}\right)^2} \quad (1)$$

Then, we averaged the instantaneous velocity over a certain period of time and obtained the averaged velocity by adopting the following equation:

$$\bar{V}_t = \frac{1}{2\tau} \int_{t-\tau}^{t+\tau} V_t dt \quad (2)$$

Hereafter, we refer the averaged velocity as the displacement rate because $\bar{V}_t$ does not mean the actual velocity in a time point but represents the amount of movement occurred within the certain period of time. We defined $\tau$ as 30 s and thus our displacement rate is equal to the amount of movement in 1-min time window. The displacement rate can be considered a general measure of movement quantity, as it captures changes in both the magnitude and frequency of movements, and it has been shown to correlate with the number of movement units[23]. Furthermore, we calculated the baseline ratio of the displacement rate as follows:

$$\text{Ratio} = \frac{\bar{V}_t}{\text{mean of } \bar{V}_t \text{ in baseline phase}} \quad (3)$$

This arrangement normalises the amount of limb movement by that occurred spontaneously in the baseline phase and focuses on the increase or decrease of limb movements relative to the baseline phase (Fig. 1b). Baseline ratio is a plausible measure to capture the variability of learning behaviours during the interaction with the mobile[28,32,40].

### Time-series clustering

We conducted the time-series clustering to elucidate the inter-individual differences of infants' learning behaviour in the mobile paradigm. This method divides the time-series data into a certain number of groups by calculating its similarities. We utilised K-means time-series clustering with Euclidian distance by adopting the following equation:

$$D = \sum_{i=1}^{k} \sum_{j=1}^{n} \sum_{t} \delta_{i,j} \left(x_{j,t} - v_{i,t}\right)^2 \quad (4)$$

Assuming that there are $n$ time-series data $\left\{x_{j,t}, | , j = 1, 2, \cdots, n\right\}$ which are to be divided into $k$ clusters. The number of clusters, $k$, is a parameter decided to capture the structure of data best. In our study, $k$ was decided as $k = 4$, and time-series data was down-sampled to 1 Hz. $D$ represents the sum of distance between each time-series data from their respective cluster centres $\left\{v_{i,t}, |, i = 1, \cdots, k\right\}$. $\delta_{i,j} = 1$ if $x_{j,t}$ is assigned to cluster $i$, and otherwise $\delta_{i,j} = 0$. The main idea of K-means clustering is the minimisation of the function $D$ by altering the assignment of each time-series data to the appropriate cluster[42]. At first, each data was randomly allocated to one of the clusters and then, iteratively, change its allocation to

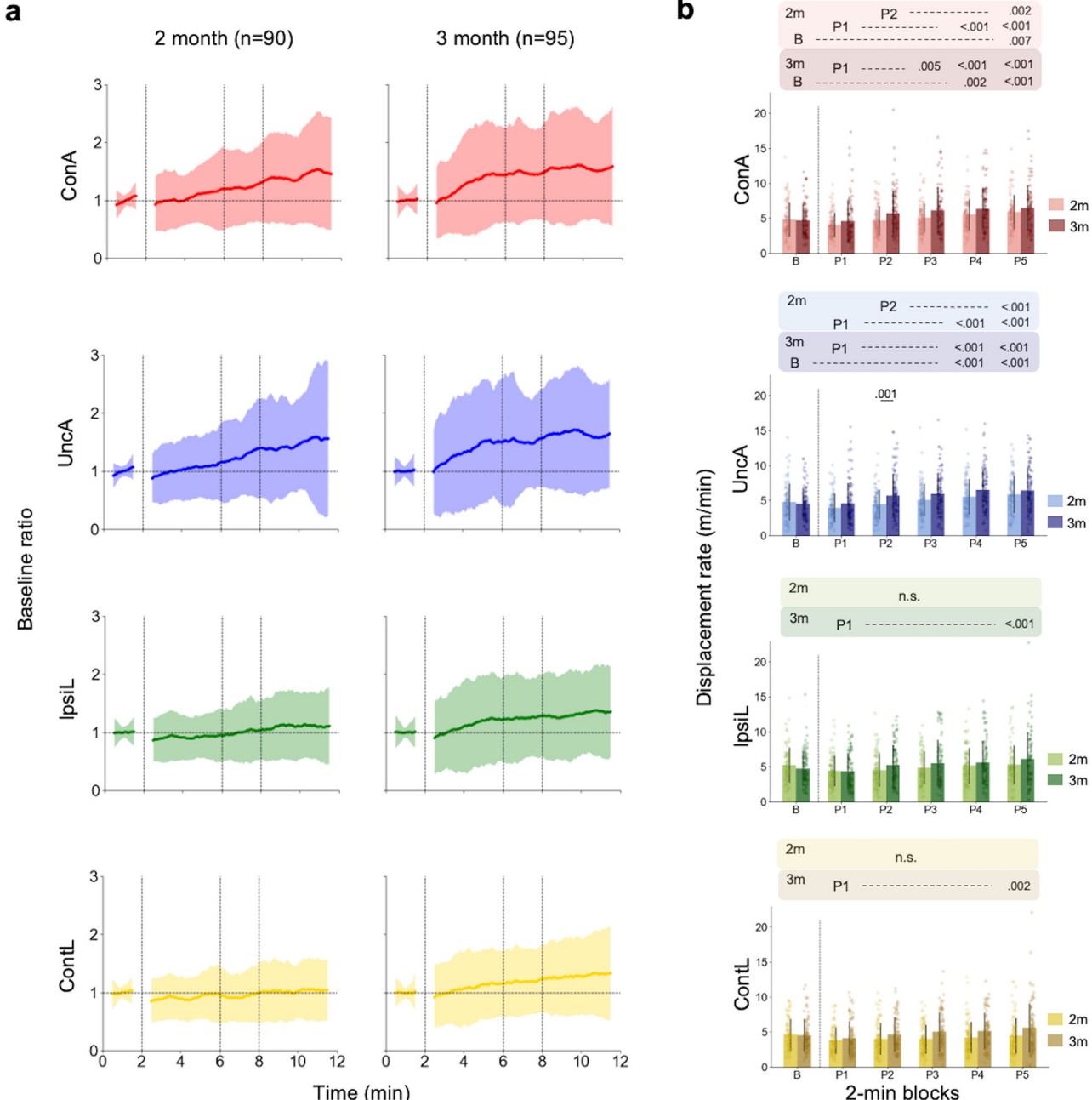

**Fig. 2 | Month-wise averaged time evolutions in movements of each limb. a** The time evolutions in baseline ratio (ConA connected arm, UncA unconnected arm, IpsiL ipsilateral leg, ContL contralateral leg). First 2 min is a baseline phase and following 10 min is a play phase. Each solid line represents the averaged time evolutions and filled area corresponds to the values between mean ± SD. **b** Month-wise mean displacement rates in every 2-min block (B Baseline phase, P Play phase, P1:

0–2 min, P2: 2–4 min, P3 4–6 min, P4 6–8 min, P5 8–10 min). Error bars show the SD of the displacement rates in each month. *p* values above each bar denote the result of the test for simple main effect of age in each phase. *p* values aligned above each panel denote the result of Tukey's HSD test for phases in each age. n.s. means that there was no statistical significance for multiple comparisons.

the cluster whose centre is nearest to the time-series data until the function $D$ cannot be reduced anymore. In our study, $x_{j,t}$ represents the time-series of baseline ratios of four limbs of $j$th infant with an assumption that baseline ratio is already scaled. In the supplementary materials, we show the results of time-series clustering when using the time series of displacement rate in the baseline phase or the play phase.

The function was implemented by using tslearn[43], a Python machine learning package for time series data. tslearn depends on basic Python packages, numpy and scipy, for array manipulation and standard linear algebra routines, and follows scikit-learn's Application Programming

Interface, which integrates a wide range of machine learning algorithms for Python.

## Statistical analysis

To reveal the statistical significance in the month-wise difference, we conducted statistical analysis on the mean displacement rate by 6 (phase: baseline, play 1, 2, 3, 4 and 5) × 2 (age: 2 and 3 month) two-way analysis of variance (ANOVA) for each limb. Phase was considered as a within-subject factor and age was considered as a between-subject factor. For multiple comparisons, we used Tukey's honestly significant difference (HSD) test. As

for the statistical significance in the cluster-wise difference, we conducted statistical analysis on the mean displacement rate of ConA by 6 (phase: baseline, play 1, 2, 3, 4 and 5) × 4 (cluster: 1, 2, 3 and 4) two-way ANOVA. Phase was considered as a within-subject factor and cluster was considered as a between-subject factor. For multiple comparisons, we used Tukey's HSD test. A two-way ANOVA was conducted using the Python packages, NumPy and pandas. Tukey's HSD test was implemented by using statmodels[44], a Python package for statistical analysis. The $p$-value threshold was 0.01 in our entire analysis. When conducting multiple tests for simple main effects, the Bonferroni method was used to correct this threshold. Data distribution was assumed to be normal, but this was not formally tested.

### Modelling
We used a dynamical systems model of mobile paradigm[36] to elucidate the relationship between spontaneous movements and learning behaviours. This model consists of three equations (Eqs. 5–7), one for the infant's limb movement, one for the mobile movement, and one for the functional coupling between the two.

$$\ddot{x} + \dot{x}\{\gamma + \alpha x^2\} + x\{\omega_0^2 + \delta x^2\} = 0 \quad (5)$$

$$\ddot{y} + \varepsilon\dot{y} + \Omega_0^2 y = cx \quad (6)$$

$$\dot{\delta} = ay^2 - \kappa\delta \quad (7)$$

Equation 5 is a van-der-Pol oscillator with an additional Duffing term, where $x(t)$ represents the infant limb movements. when $\gamma < 0$ and $\alpha > 0$, $x(t)$ falls into a limit cycle and exhibit periodic movements spontaneously. $\omega_0$ determines the natural oscillation frequency and $\delta x^2$ modifies it during the interaction with the mobile. $\delta(t)$, a variable in Eq. 6, increases the limb oscillation frequency when the mobile moves. Equation 7 is a damped harmonic oscillator, where $y(t)$ represents the mobile movement. $\varepsilon$ is a damping term and $\Omega_0$ determines the eigenfrequency. The connection between infant and mobile is determined by the value of $c$. If $c = 0$, the mobile oscillator is disconnected from the infant limb oscillator. This setting regards as a baseline phase. If $c > 0$, the mobile oscillator is forced by the infant limb oscillator. This setting regards as a play phase. Equation 7 expresses the interaction between $x(t)$ and $y(t)$. $a$ is a key parameter for behavioural changes because it serves to couple the infant and the mobile, which also determines the strength of positive feedback regarding the activation of limb movement. $\kappa$ limits the increase when $\delta(t)$ is enough high. Details of this model and its behaviour are described in Kelso and Fuchs[36]. For our simulations, the following parameters were used: $\gamma = -0.25, \alpha = 1, \varepsilon = 1, \Omega_0 = 2.2, a = 0.13, \kappa = 0.007$. $c = 0$ for baseline phase and $c = 2$ for play phase. $\omega_0$ was generated from uniform distributions whose range was 0.5 and mean value was determined for each cluster. The differential equations for the model were integrated numerically with a time step of 0.01 using Odeint, a scipy library for numerical simulations.

### Reporting summary
Further information on research design is available in the Nature Portfolio Reporting Summary linked to this article.

## Results
### Conventional group averaged learning curve and individual differences
185 infants (2-month-old: $n = 90$, 3-month-old: $n = 95$) completed the experimental procedure, which consists of 2-min baseline phase and 10-min play phase. Displacement rates were calculated to measure the amount of movement in each limb within a 1-min time window. This measure reflects the extent of limb movement during a given time window, while its ratio to the baseline level (baseline ratio) indicates the degree of increase or decrease in limb movement relative to the baseline phase (see 'Methods' for a detailed description).

We first used a conventional approach to test age-dependent differences in group-averaged learning curves to infer age-dependent maturation of learning capabilities[23,31,40,45]. The learning curve was calculated as a group-averaged time evolution in the baseline ratio[23,28,40]. The averaged learning curve for the arm connected to the mobile (ConA) showed a general increase over time in both 2 and 3-month-old infants, as shown in Fig. 2a. Changes were also observed in the limbs not connected to the mobile. To clarify the characteristics of learning curves for each age group, we conducted statistical analysis on the mean displacement rate by 6 (phase: baseline, play 1, 2, 3, 4 and 5) × 2 (age: 2 and 3 months) analysis of variance (ANOVA). The mean displacement rate indicates the averaged amount of the limb movement in 1-min time window for each phase. Phase was considered as a within-subject factor and age was considered as a between-subject factor. As for the connected arm (ConA), significant main effects were observed for the phases ($F(5, 915) = 42.90$, $p < 0.001$, $\eta_p^2 = 0.190$, 95% CI = [0.144, 0.231]), but not for the age ($F(1, 183) = 3.86$, $p = 0.051$, $\eta_p^2 = 0.021$, 95% CI = [0.0, 0.078]). In addition, significant interaction between the ages and phases was observed ($F(5, 915) = 3.95$, $p = 0.002$, $\eta_p^2 = 0.021$, 95% CI = [0.003, 0.038]). With regard to the interaction, there were significant simple main effects of the phases in both age: 2-month-old ($F(5, 915) = 17.55$, $p < 0.001$, $\eta_p^2 = 0.088$, 95% CI = [0.052, 0.120]) and 3-month-old ($F(5, 915) = 29.30$, $p < 0.001$, $\eta_p^2 = 0.138$, 95% CI = [0.096, 0.176]). For a multiple comparison, the Tukey's HSD tests were conducted (Fig. 2b). The details of the multiple comparisons are provided in the Supplementary Tables 1 and 2. In comparison to the baseline phase, 2-month-old infants increased their movements in the connected arm only during the last 2 min of the play phase (P5), whereas 3-month-old infants increased them from 6 min after the mobile connection (P4). Significant differences were also observed between P1-P4, P1-P5 and P2-P5 in 2-month-old infants; and P1-P3, P1-P4 and P1-P5 in 3-month-old infants. For the unconnected arm (UncA), significant main effects were observed for the phases ($F(5, 915) = 38.94$, $p < 0.001$, $\eta_p^2 = 0.175$, 95% CI = [0.131, 0.216]), but not for the age ($F(1, 183) = 4.34$, $p = 0.039$, $\eta_p^2 = 0.023$, 95% CI = [0.0, 0.082]). In addition, significant interaction between the age and phases was observed ($F(5, 915) = 4.64$, $p < 0.001$, $\eta_p^2 = 0.025$, 95% CI = [0.005, 0.043]). With regard to the interaction, there was a significant simple main effect of the age in play 2 ($F(1, 183) = 10.95$, $p = 0.001$, $\eta_p^2 = 0.056$, 95% CI = [0.009, 0.133]). With respect to the effect of phases, there were significant simple main effects of the phases in both age: 2-month-old ($F(5, 915) = 17.30$, $p < 0.001$, $\eta_p^2 = 0.086$, 95% CI = [0.051, 0.119]) and 3-month-old ($F(5, 915) = 26.28$, $p < 0.001$, $\eta_p^2 = 0.126$, 95% CI = [0.085, 0.163]). The results of Tukey's HSD test show that there were statistical significances between P1-P4, P1-P5, and P2-P5 in 2-month-old infants; and B-P4, B-P5, P1-P4, and P1-P5 in 3-month-old infants. The details of the multiple comparisons are provided in the Supplementary Tables 3 and 4. For the ipsilateral leg (IpsiL), significant main effects were observed for the phases ($F(5, 915) = 15.85$, $p < 0.001$, $\eta_p^2 = 0.080$, 95% CI = [0.046, 0.111]), but not for the age ($F(1, 183) = 0.92$, $p = 0.339$, $\eta_p^2 = 0.005$, 95% CI = [0.0, 0.044]). In addition, significant interaction between the age and phases was observed ($F(5, 915) = 4.70$, $p < 0.001$, $\eta_p^2 = 0.025$, 95% CI = [0.006, 0.044]). With regard to the interaction, there were significant simple main effects of the phases in both age: 2-month-old ($F(5, 915) = 5.65$, $p < 0.001$, $\eta_p^2 = 0.030$, 95% CI = [0.009, 0.050]) and 3-month-old ($F(5, 915) = 14.90$, $p < 0.001$, $\eta_p^2 = 0.075$, 95% CI = [0.042, 0.106]). In the Tukey's HSD test, the significant difference was observed only between P1 and P5 in 3-month-old infants. The details of the multiple comparisons are provided in the Supplementary Tables 5 and 6. For the contralateral leg (ContL), significant main effects were observed for the phases ($F(5, 915) = 12.22$, $p < 0.001$, $\eta_p^2 = 0.063$, 95% CI = [0.032, 0.091]), but not for the age ($F(1, 183) = 4.64$, $p = 0.033$, $\eta_p^2 = 0.025$, 95% CI = [0.0, 0.085]). In addition, significant interaction between the age and phases was observed ($F(5, 915) = 5.30$, $p < 0.001$, $\eta_p^2 = 0.028$, 95% CI = [0.008, 0.048]). With regard to the interaction, there were significant simple main effects of the phases in both age: 2-month-old ($F(5, 915) = 4.69$, $p < 0.001$, $\eta_p^2 = 0.025$, 95% CI = [0.006, 0.044]) and 3-month-old ($F(5, 915) = 12.83$, $p < 0.001$, $\eta_p^2 = 0.066$, 95% CI = [0.034, 0.095]). In the Tukey's HSD test, the significant

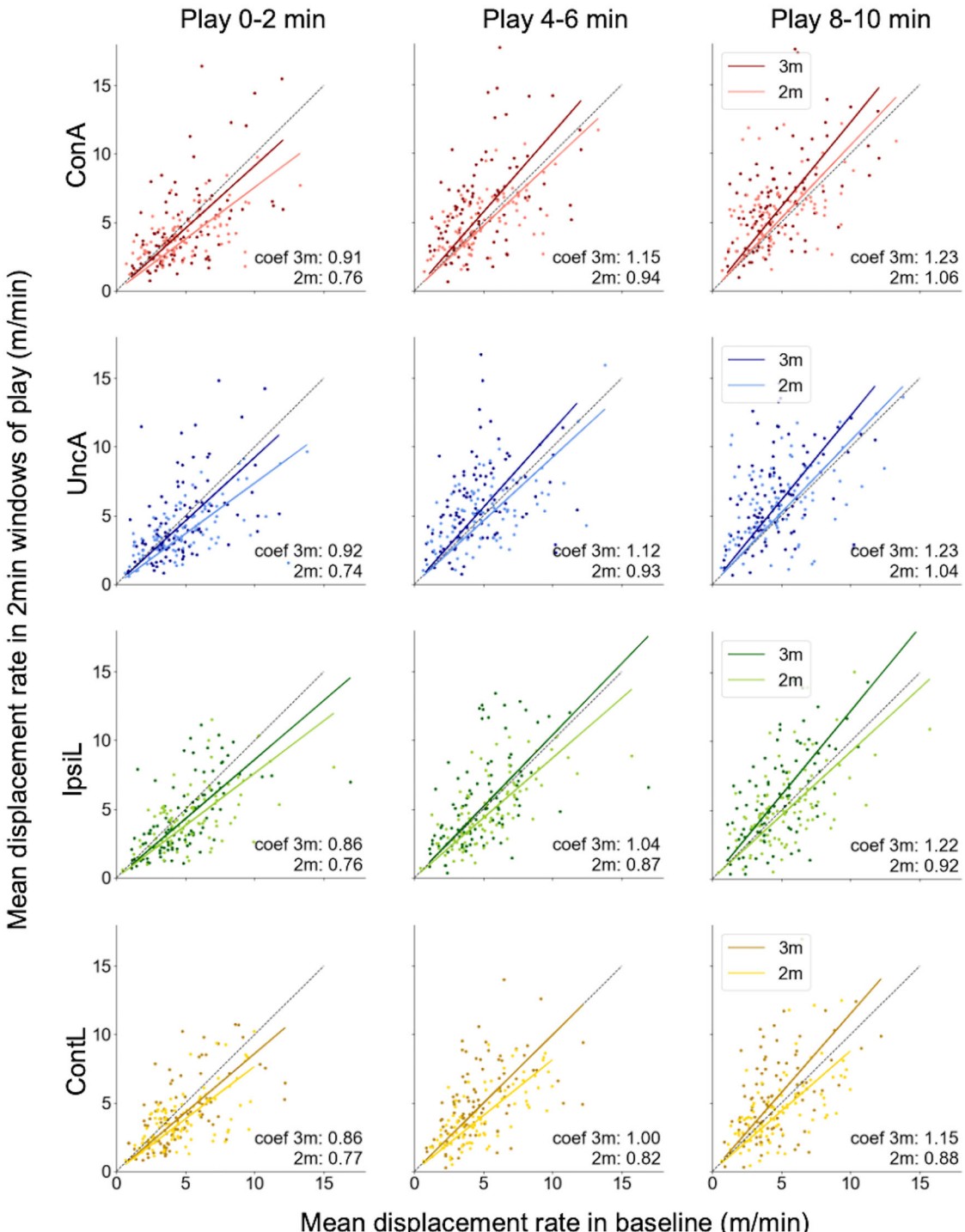

**Fig. 3 | Scatter plot for course-grained time evolutions of displacement rates coloured by limb and age.** *X*-axis is the mean displacement rate in baseline phase. *Y*-axis is the mean displacement rate in 2-min windows of play phase: first 2 min (left), middle 2 min (centre), and last 2 min (right). A dark-coloured dot represents one 3-month-old infant and a light-coloured dot represents one 2-month-old infant (ConA connected arm, UncA unconnected arm, IpsiL ipsilateral leg, ContL contralateral leg). The dashed line of $y = x$ is the boundary between the increase or decrease of limb movements compared to the baseline level. 'coef' in each panel represents the coefficient of regression lines for each age group.

difference was observed only between P1 and P5 in 3-month-old infants. The details of the multiple comparisons are provided in the Supplementary Tables 7 and 8.

Applying the conventional group-averaging analysis, we confirmed that both 2 and 3-month-old infants increased their arm movements as a result of the mobile connection. In particular, 3-month-old infants increased movements in both the connected and unconnected arms during the earlier phases, whereas 2-month-old infants increased movements in the connected arm only during the final P5 phase, suggesting that there were age-related differences in the timing of movement changes. However, significant differences in movements of the connected arm were not observed between the ages in either the baseline and play phases, suggesting that data from different age groups included characteristics that showed similar changes. Furthermore, there is a large magnitude of standard deviations in

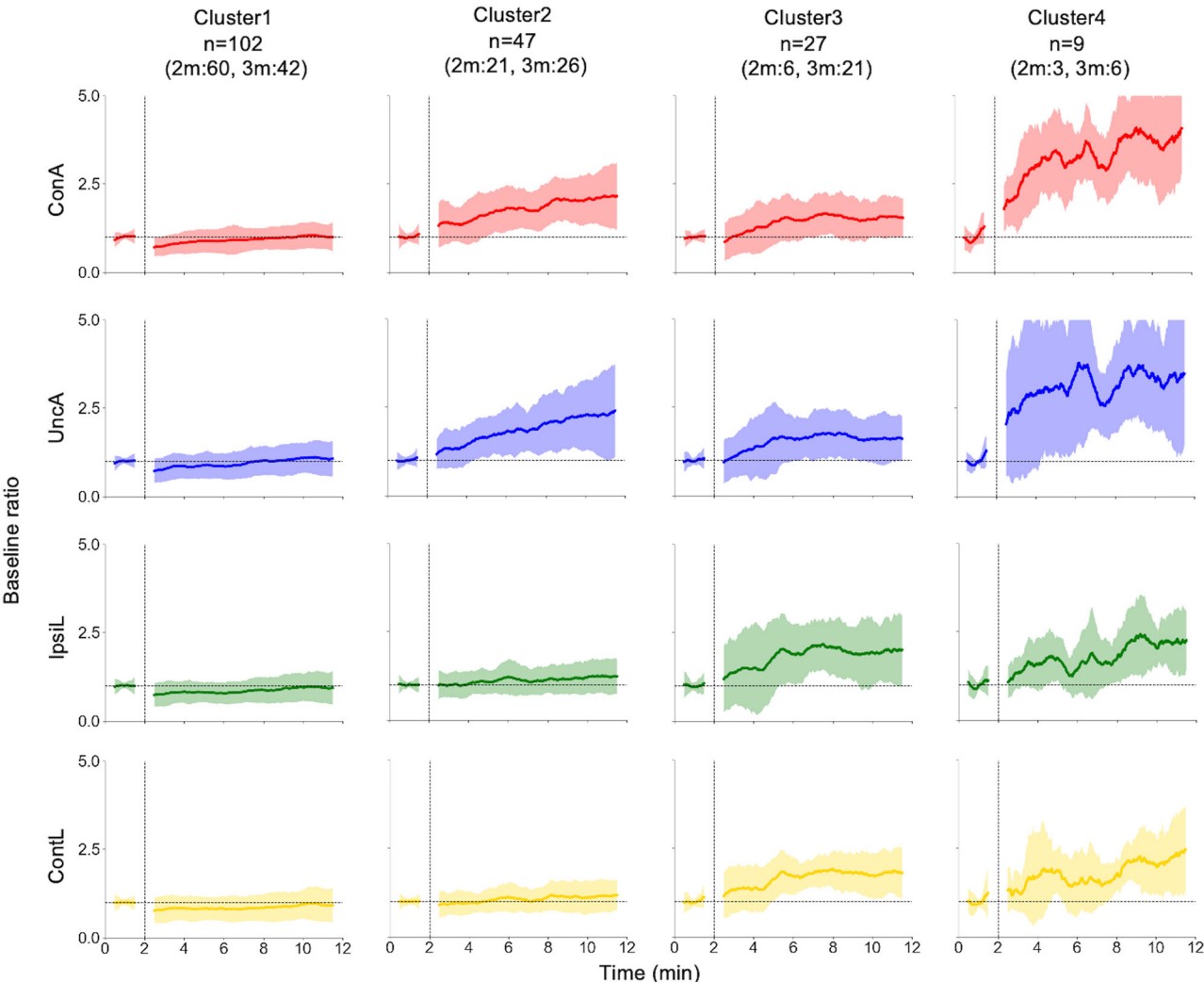

**Fig. 4 | Cluster-averaged time evolutions of baseline ratio coloured by limb.** First 2 min is a baseline phase and following 10 min is a play phase. ConA connected arm, UncA unconnected arm, IpsiL ipsilateral leg, ContL contralateral leg. 102 infants were assigned to cluster 1, in which the averaged baseline ratio of all limbs did not increase. Forty-seven infants were assigned to cluster 2, in which the averaged baseline ratio of both arms increased. Twenty-seven infants were assigned to cluster 3, in which the averaged baseline ratio of all limbs, relatively, both legs increased. Nine infants were assigned to cluster 4 in which the averaged baseline ratio of both arms increased more than cluster 2 or cluster 3. The averaged baseline ratio of both legs also increased in the cluster 4.

all panels, which indicates that age differences alone cannot reveal the covert large inter-individual differences in behavioural changes.

To disclose the individual differences in the change of limb movements, we scattered the 2-min-averaged displacement rate in the play phase as a function of that in the baseline phase, as shown in Fig. 3.

In each panel, a dark-coloured dot represents a 3-month-old infant, while a light-coloured dot represents a 2-month-old infant. The dashed line of $y = x$ is the boundary between the increase or decrease of limb movements. If dots are in the region of $y > x$, the amount of limb movement is greater than that in the baseline phase (baseline ratio >1). If dots are in the region of $y < x$, the amount of limb movement is smaller than that in the baseline phase (baseline ratio <1). The slope of the line passing through the origin and each dot corresponds to the baseline ratio for each dot. The coloured lines represent the regression lines passing origin for each age group. While the regression lines indicate that 3-month-old infants increase their limb movements more than 2-month-old infants in all panels, the distribution of the data for each age group considerably overlaps. Thus, the age difference cannot fully explain the individual differences in limb movement alterations. As shown in Fig. 3, there are large inter-individual differences in the displacement rates during both baseline and play phases, as well as in the baseline ratio. This suggests that both the initial states,

reflecting spontaneous movements before learning, and the final states, representing changed movements after learning, showed significant individual variability in both 2- and 3-month-old infants.

### Analysis of individual differences by clustering
In addition to individual differences in initial and final states, individual learning curves in our data likely involve not only gradual increases but also various dynamic patterns. To uncover individual differences in the learning process, it is essential to analyse the entire dynamics of behavioural trajectories, including early-phase changes. Furthermore, it is expected that diverse changes in movement patterns occur on an individual basis, including not only the limb connected to the mobile but also the movements observed in the other limbs. To comprehensively classify distinct dynamic patterns, we applied time-series clustering[46] to the combined data of 2- and 3-month-old infants.

By a clustering analysis of the time evolution of baseline ratios, infants were divided into four clusters: No increase (cluster 1), Arm increase (cluster 2), All limb increase (cluster 3), and Arm more increase (cluster 4), as shown in Fig. 4. Each cluster has a certain number of 2- and 3-month-old infants.

Cluster 1 contains the largest number of infants ($n = 102$, 55% of the participants), who did not increase their limb movements with the mobile

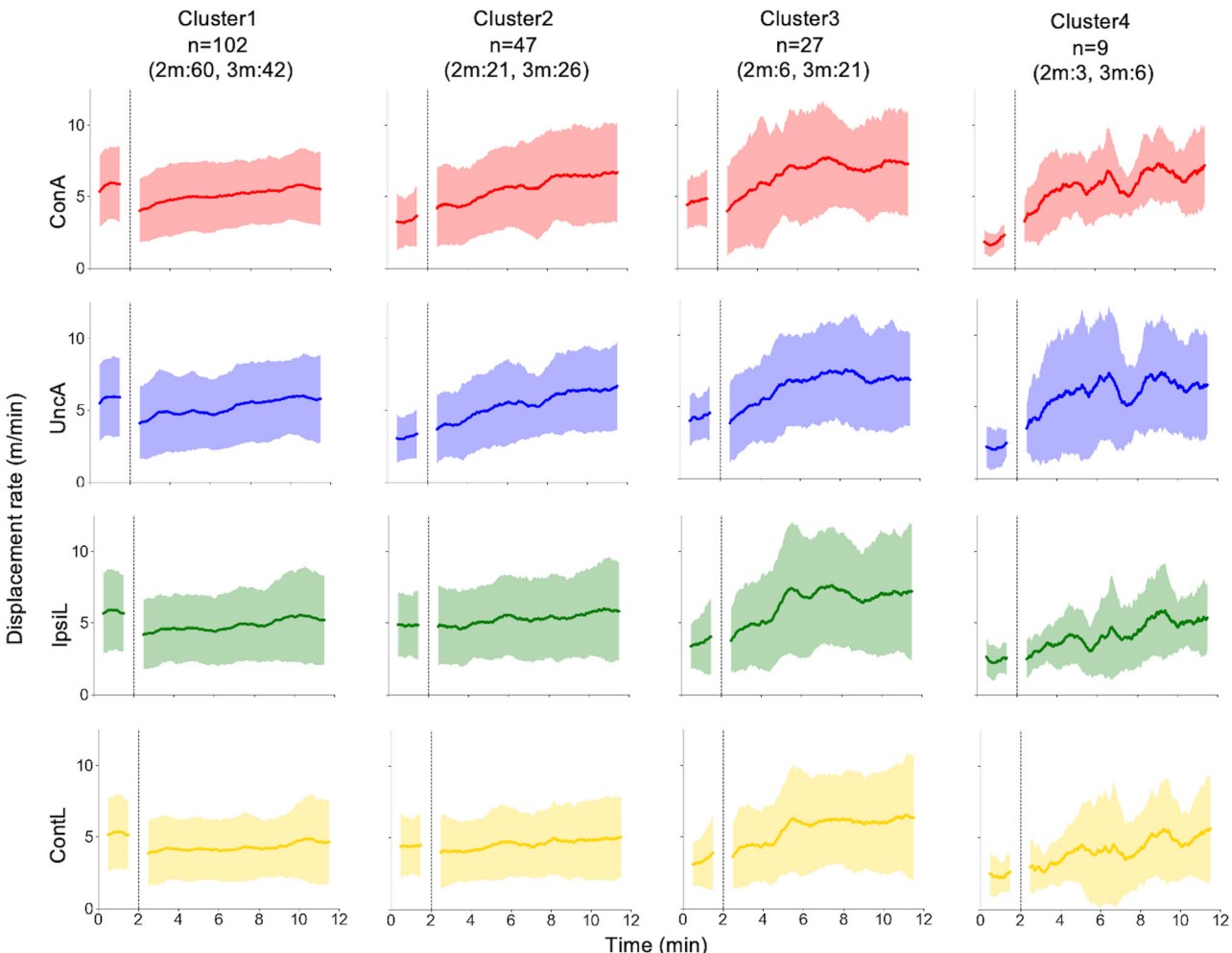

**Fig. 5 | Cluster-averaged time evolutions of displacement rate coloured by limb.** ConA connected arm, UncA unconnected arm, IpsiL ipsilateral leg, ContL contralateral leg. The difference between clusters in displacement rates is observed in the baseline phase rather than the play phase. The displacement rate in the baseline phase is low when the correspondent baseline ratio is high.

connection. Two-thirds of the 2-month-old infants and more than one-third of the 3-month-old infants belong to this cluster. Cluster 2 includes one fourth of the participants, who increased both arm's movements to about twice the baseline level. Cluster 3 comprises about 15% of the participants, who increased movements of all limbs, including the legs. The number of 3-month-old infants was relatively larger than that of 2-month-old infants in this cluster. Cluster 4 consists of a few infants who increased their limb movements much more than those in other clusters, particularly increasing their arm movement to about three or four times the baseline level. These results indicate that the degree of limb movement alterations is varied across individuals and it is not explained by age differences. It is worth noting that more than half of the infants did not increase their limb movements.

Figure 4 illustrates the diversity in patterns of limb movement alterations; however, they were normalised to the amount of limb movement in the baseline phase. To examine whether the variability in spontaneous movements at baseline prior to learning affects the results within each cluster, we analysed cluster-wise time evolutions of displacement rate for each limb without normalising by the baseline as shown in Fig. 5. Specifically, infants in Cluster 1, characterised by no increase, had high baseline values, experienced a temporary decrease at P1, and then reached similar values to those of infants in other clusters by P5. Notably, the cluster-wise difference in displacement rates was observed in the baseline phase rather than in the play phase.

To confirm these characteristics, we conducted statistical analysis on the mean displacement rate of ConA by 6 (phase: baseline, play 1, 2, 3, 4, and 5) × 4 (cluster: 1, 2, 3, and 4) ANOVA (Fig. 6). Phase was considered as a within-subject factor and cluster was considered as a between-subject factor. Significant main effects were observed for the phases ($F(5, 905) = 57.14$, $p < 0.001$, $\eta_p^2 = 0.240$, 95% CI = [0.192, 0.283]), but not for the clusters ($F(3, 181) = 1.196$, $p = 0.313$, $\eta_p^2 = 0.019$, 95% CI = [0.0, 0.062]). In addition, significant interaction between the phases and clusters was observed ($F(5, 905) = 22.87$, $p < 0.001$, $\eta_p^2 = 0.112$, 95% CI = [0.073, 0.148]). With regard to the interaction, there were significant simple main effects of the phases in cluster 2 ($F(5, 905) = 18.38$, $p < 0.001$, $\eta_p^2 = 0.092$, 95% CI = [0.056, 0.126]), cluster 3 ($F(5, 905) = 19.81$, $p < 0.001$, $\eta_p^2 = 0.097$, 95% CI = [0.062, 0.133]), and cluster 4 ($F(5, 905) = 38.75$, $p < 0.001$, $\eta_p^2 = 0.176$, 95% CI = [0.131, 0.217]). With respect to the effect of clusters, there was a significant simple main effect of the clusters in baseline ($F(3, 181) = 34.82$, $p < 0.001$, $\eta_p^2 = 0.366$, 95% CI = [0.255, 0.457]), play 3 ($F(3, 181) = 8.82$, $p < 0.001$, $\eta_p^2 = 0.128$, 95% CI = [0.043, 0.213]), play 4 ($F(3, 181) = 5.57$, $p = 0.001$, $\eta_p^2 = 0.084$, 95% CI = [0.016, 0.161]). The results of Tukey's HSD test are described in Fig. 6. The details of the multiple comparisons are provided in the Supplementary Tables 9–18. The statistical analyses for other limbs are shown in Figs. S1–S3 and Supplementary Tables 19–48.

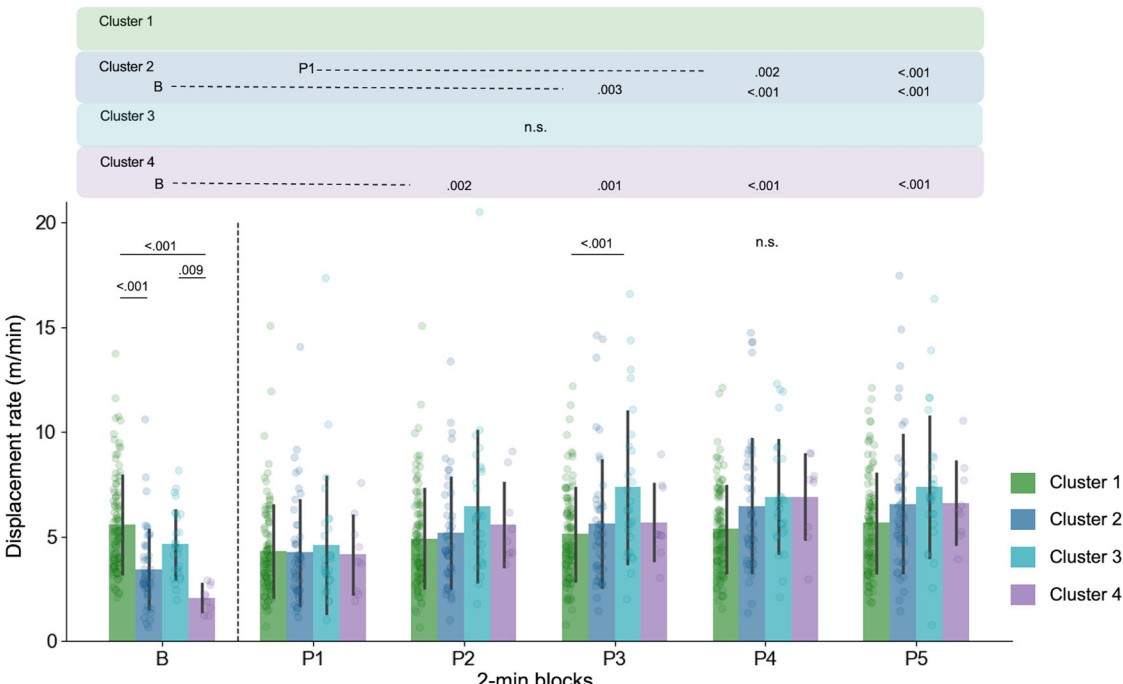

**Fig. 6 | Cluster-wise mean displacement rate of ConA.** Mean displacement rates in each phase are shown (B Baseline phase, P Play phase, P1: 0–2 min, P2: 2–4 min, P3: 4–6 min, P4: 6–8 min, P5: 8–10 min). Error bars show the SD of the displacement rates in each cluster. *p* values above each bar denote the result of Tukey's HSD test for clusters in each phase. *p* values aligned above the panel denote the result of Tukey's HSD test for phases in each cluster. n.s. means that there was no statistical significance for multiple comparisons. We did not conduct the multiple comparisons in cluster 1, P1, P2 and P5 because the simple main effects of phases or clusters were not significant.

The inter-cluster differences in the displacement rate were observed in the baseline phase and the P3 phase. Infants in cluster 2 and 4 exhibited significantly lower amount of spontaneous movements than cluster 1 in the baseline, and a significant increase in movements was only observed in these two clusters. Significant increases were observed between B-P3, B-P4, B-P5, P1-P4 and P1-P5 in cluster 2; and B-P2, B-P3, B-P4 and B-P5 in cluster 4. It is worth noting that during the play phase, the significant inter-cluster difference in the displacement rate was only observed in the P3 phase. The inter-cluster differences in the baseline phase disappeared at the P1 phase by the increase (cluster 2 and 4) and the decrease (cluster 1) in the amount of limb movement. Behavioural changes in the early play phase became apparent through this clustering analysis. These results suggest that infants' limb movement alteration patterns are largely determined by the amount of spontaneous movements (see also Figs. S1–S3). In other words, the individual trajectories of behavioural change evolved from a state unique to each cluster to one indistinguishable from others. Environmental interactions serve to produce similar patterns with reduced individual differences.

Additionally, we created illustrations of the time evolutions of baseline ratio and displacement ratio for all limbs using individual data, sorted by clusters and ages without averaging, to highlight individual differences as shown in Fig. S4. As anticipated from the previous analyses, the curves of the baseline ratio were clustered based on the relative differences in behaviours during the play phase compared to the baseline, while the curves of the displacement rate primarily reflected baseline differences between clusters, making distinctions in the play phase less apparent.

The cluster analysis elucidated nomothetic characteristics of individual differences; however, individual behaviours were far more complex. The infant-wise four-limbs dynamics for the baseline ratio and the displacement ratio are shown in Figs. S5 and S6, respectively. The curves showed not only a gradual and monotonic increase but also decreases and fluctuations. Notably, many infants exhibited rhythmic cycles of increases and decreases during the play phases. Each of these dynamic changes may reflect behavioural fluctuations of individual infants with different timescales. It is evident from the individual data that curves exhibiting a monotonic increase, as expected in an ideal learning curve, were, in fact, quite rare. Additionally, individual data from infants in Cluster 4 showed a high baseline ratio (Fig. S5), while the actual amount of movement, measured by displacement rate (Fig. S6), was not much different from other clusters during the play phase. This also indicates that they exhibited a lower amount of spontaneous movement in the individual level.

**Dynamical systems model explaining individual differences**
In the previous section, we have demonstrated that infants can be categorised into four clusters based on the time evolution of limbs movement patterns. The differences between each cluster largely depended on the amount of spontaneous movements at the baseline phase. To elucidate a possible mechanism by which spontaneous movements influence behavioural changes during learning, we attempted to replicate the diverse time evolution of limb movement changes using simulations based on a dynamic systems model of the mobile paradigm[36] (see 'Methods' for a detailed description).

The Eq. (5) is a van-der-Pol oscillator, which corresponds to infant's limb movements. The Eq. (6) is a damped harmonic oscillator, which represents mobile movements forced by the limb movements. The Eq. (7) creates a positive feedback loop between infant's limb movements and mobile movements: the motion of the mobile induces more vigorous limb movements, which further produce more intense mobile movements. The parameter $a$ was defined as a key parameter determining the degree of infants' learning capabilities[36]. The differences in the amount of spontaneous movement can be represented by the natural oscillation frequency ($\omega_0$) in the infant limb movement dynamics. Thus, we assumed that each infant had a different value of $\omega_0$, determined by fitting it to the experimental data, while the parameter $a$ for learning remained the same. Then, the time evolutions in baseline ratios of ConA were simulated as shown in Fig. 7.

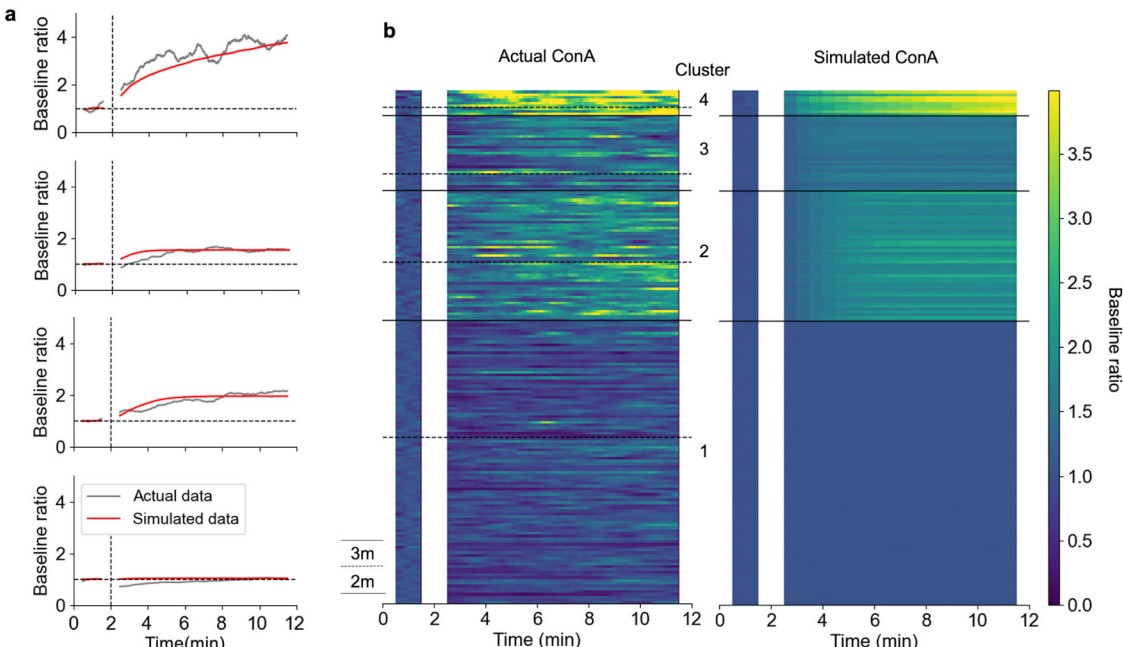

**Fig. 7 | Simulated time evolutions in baseline ratios of ConA. a** Actual time evolutions (grey) and its simulation (red) for cluster 1, 2, 3, and 4 arranged from bottom to top. The natural oscillation frequencies were chosen from a uniform distribution whose range was $2.8 < \omega_0 < 3.3$ (Cluster 1), $0.9 < \omega_0 < 1.4$ (Cluster 2), $1.4 < \omega_0 < 1.9$ (Cluster 3), and $0.3 < \omega_0 < 0.8$ (Cluster 4). Simulations were conducted for the same number as each cluster size and simulated time evolutions of baseline ratio were averaged for each cluster. **b** Infant-wise plot of actual and simulated time evolutions in baseline ratio for connected arm (ConA). Dashed lines in the actual data divides 2- or 3-month-old infants within a cluster.

In every simulation, $\omega_0$ was generated from a uniform distribution for each infant in each cluster whose range was $2.8 < \omega_0 < 3.3$ (Cluster 1), $0.9 < \omega_0 < 1.4$ (Cluster 2), $1.4 < \omega_0 < 1.9$ (Cluster 3) and $0.3 < \omega_0 < 0.8$ (Cluster 4). The simulated cluster-averaged time evolution of the baseline ratio of ConA were similar to the experimental results. This suggests that the variation in intrinsic dynamics for spontaneous movements, despite identical learning capabilities, was sufficient to account for the distinct patterns of behavioural changes.

## Discussion

We investigated the individual differences in how young infants change their limb movements during the interaction with an overhead mobile to acquire instrumental actions. The results are summarised as follows. (1) The averaged learning curve for the arm connected to the mobile showed a general increase over time in both 2 and 3-month-old infants, confirming the evidence of learning at the group level. In addition, 3-month-old infants showed an earlier increase in arm movements during the play phase compared to 2-month-old infants at the group level. Focusing on the variability of the data, both the initial state, reflecting spontaneous movements before learning, and the final states, representing instrumental movements after learning, exhibited significant individual differences in both 2-month-old and 3-month-old infants. (2) By a clustering analysis of the time evolution of baseline ratio, infants were divided into four clusters: no increase, arm increase, all limb increase, and arm more increase. Each cluster included infants from both age groups, and the differences between clusters could not be explained by age-dependent differences in behavioural changes. Importantly, the differences in clusters were marked during the baseline but were not evident by the end of the play phase. Viewed as a group, individual trajectories of behavioural change evolved from a state unique to each cluster to one indistinguishable from others after learning. Furthermore, even within the same cluster, individual trajectories exhibited complex changes over a shorter time scale. (3) A dynamical systems model showed that the variation in intrinsic dynamics for spontaneous movements, despite identical learning capabilities, was sufficient to account for the distinct patterns of behavioural changes. Overall, these findings reveal an aspect of learning as a process that reduces individual differences.

This study demonstrates that in analysing infant learning, it is essential to take into account not only the differences before and after learning but also the state prior to learning. Traditionally, the criterion for establishing learning has been defined as reaching a baseline ratio of 1.5[3,29,30]. Unlike many previous studies that measured the number of kicks by attaching a string to the leg, this study attached the string to the arm and used a motion analysis system to measure movement[23]. Nevertheless, if the traditional criterion for learning is applied, it would mean that the infants in Cluster 1 of this study—representing more than half of all participants—did not achieve learning. However, these infants exhibited unique characteristics, including high levels of spontaneous movements before learning and a temporary decrease in movement immediately after the start of the play phase. This suggests the presence of mechanisms, such as increased attention to the environment and motor inhibition[38,40]. A previous research has also reported that infants with high initial spontaneous movement frequencies are less likely to achieve high baseline ratios[32,47]. From a dissenting perspective, one might doubt that our results could be explained by a ceiling effect, which arises when a test has an upper limit and the collected data are distributed near that limit. However, there were substantial fluctuations in movements during the play phase, which cannot be explained by a simple model where learning increases monotonically until a ceiling is reached. For these infants, the mobile may already move sufficiently due to their limb movements, eliminating the need to increase activity further. Instead, they may be learning that reducing their limb movements results in less mobile motion[4]. Consequently, it is inappropriate to conclude that infants in Cluster 1 failed to learn. In the context of the mobile paradigm, an extinction phase has been used as a procedure to examine whether behaviour is generated based on memory by measuring baseline ratio or retention ratio[22,48]. However, this procedure has its limitations. One major weakness is that learning in a certain number of infants cannot be assessed using the ratio in the amount of movement. Additionally, some infants may quickly realise

that moving their arms no longer causes the mobile to move, and consequently stop the behaviour they had previously learned and generate another behavioural change. In other words, it cannot be ruled out that a new learning process may begin under these altered environmental conditions. In the present study, our focus was to elucidate the diverse behaviours exhibited by infants to learn the contingency between their own movements and the environmental changes during a sufficiently long play phase. Therefore, we did not include an extinction phase. While classical studies on memory using the mobile paradigm have provided foundational insights[22,48], a recent theoretical development has begun to shed new light on the dynamics of memory[15] and future research should include finer-grained analyses of movement changes on shorter timescales[4] and additional internal state measures such as EEG[49,50].

It has been shown that behaviours of infants, including movement[16,26,51] and perception[52], undergo dramatic changes between 2 and 3 months of age. The ability to distinguish whether the movement of the mobile is caused by one's own actions or by others develops between 2 and 3 months of age[40]. Furthermore, the pattern of brain activity in response to the audiovisual stimuli generated by the mobile's movement undergoes significant changes between 2 and 3 months of age[53,54]. In the context of the mobile paradigm, many studies have focused on this age range. However, the definition of age groups varies across studies. For instance, some studies define the 3-month age group as uniformly distributed across 30 days, while others adopt narrower age ranges, potentially leading to different outcomes. A previous study reported that the characteristics of movement changes during mobile tasks in 3- to 4-month-old infants vary depending on age groups defined with a 10-day range[31]. In the present study, we aimed to dissociate age-related differences from individual differences by setting up groups of 2- and 3-month-old infants with precise age distributions spanning 10 days. At the group level, we observed age-related differences in average behaviours. Nonetheless, at the individual level, much of the data could not be fully explained by the difference in age: each cluster included a certain number of both 2- and 3-month-old infants. This finding suggests substantial individual differences in behavioural changes at both ages.

The reproducibility challenges in behavioural research are influenced by factors such as sample size and statistical methodologies. The number of infants who participated in this study ($n = 279$), and the number included in the final analysis ($n = 185$), represent a considerably large sample size compared to those typically used in the mobile paradigm, as this research focuses on individual differences. The sample size also exceeds the average sample size of 108.92 in general studies on individual differences in infancy[39]. The statistical significance threshold for group analysis in the present study was set at $p = 0.01$[55]. Additionally, the reproducibility challenges may be closely associated with individual differences inherent in subjects, particularly infants. For instance, there is an issue of reproducibility on whether arm- and leg-movement differentiation during mobile task exists in early infancy[3]. Some studies reported that limb movement differentiation is observed in 3 or 4 months of age[23,27,45], while other studies showed no such evidence[3,56]. This inconsistency is probably related to the individual differences in limb movement alteration patterns. Our clustering analysis showed that two-third of 3-month-old infants did not differentiate their arm and leg movements (cluster 1 and 3), whereas remaining one-third of 3-month-old infants (cluster 2 and 4) differentiated them. The inter-cluster differences are related to whether there is a bias in the amount of spontaneous movement between the arms and legs. Taking into account the individual differences in spontaneous movement patterns, it is suggested that the process of learning to move the mobile involves both age-related differences and inter-individual differences, and the behavioural changes related to learning are not necessarily limited to the connected limb alone. Recruiting a sufficient number

of infants within a restricted age range and extracting subgroups based on the individuality of infants' spontaneous movements could lead to more reproducible analytical results. Understanding the dynamic properties of behavioural changes will lead to identify nomothetic principles applicable across individual learning trajectories and bring an alternative perspective on the issue of reproducibility. Clustering analyses with different tasks and different age groups have also revealed various learning trajectories at the individual level hidden under the averaged description, such as hand preference from birth to 24 months[8], and maturation of walking skills during a several months from the onset of independent upright locomotion[13]. Consistent with our findings, subgroup analyses in these studies elucidated that initially diverse states gradually converged into similar patterns with reduced individual differences through interactions with the environment. This convergence is considered a nomothetic phenomenon in learning processes, and the present study further demonstrated that such a shift can emerge even within a brief timescale of approximately 10 min.

The mobile paradigm is one of the few experimental setups applicable in early infancy to elucidate the processes through which humans interact with their environment for the first time, adapt their behaviour, and establish learning and memory[1,4,22–25]. As evidenced by the data from this study, the averaged movement changes in infants at the group level increase over time during playing. This trend often leads to interpreting the learning observed in this paradigm as simple reinforcement learning, where behaviours monotonically increase in frequency in response to stimuli associated with their actions[57]. However, a detailed analysis of movement changes at the individual level revealed the complex patterns that cannot be fully explained by the reinforcement alone. Notably, there were substantial individual differences in spontaneous movements prior to learning. A recent study demonstrated that spontaneous movements generate sensorimotor coordination even in the absence of specific goals toward the environment[58], contributing to the formation of unique movement attractors in each infant. Given that the spontaneous formation of these attractors occurs without a predefined direction, it is likely to exhibit considerable diversity. However, once interaction with the environment begins, the movement dynamics are modified to adapt to environmental contingencies, potentially leading to a reduction of individual differences. The emergence of rhythmic and stereotypical movements during this developmental period[59–61] may provide opportunities for sensorimotor adaptation by promoting interaction with the environment through the use of available motor repertoires of spontaneous movements[23]. This suggests that individual behavioural histories may reflect the emergence of diverse mechanisms during interactions with the environment. These mechanisms may include exploratory behaviours during interaction[20], a sense of agency[4,40,49], preparation for goal-directed actions[51], and formation of long-term memory[15,22]. The integration of these mechanisms constitutes the dynamics of individual motor behaviour, and serve as a determinant of both the selection of behavioural strategies[9] and the degree of movement maturity[10] across development.

## Limitations

In the present study, we focused on the individual differences in the quantity of limb movements. However, a qualitative analysis of movement trajectories may further elucidate individual differences in infants' behavioural changes. Dimensionality reduction techniques such as PCA, as well as more recently developed methods like UMAP[62] and t-SNE[63], may be useful for exploring the dynamic evolution of qualitative patterns. In addition, limb synergies have been reported as effective for analysing the patterns of limb movements[41]. Future studies should investigate individual differences in the qualitative patterns of infants' behavioural changes from spontaneous to instrumental using these techniques, leading to provide further support for the idea that the patterns of spontaneous movements are closely linked to learning.

Additionally, in our measurement, the infants were placed in an unfamiliar laboratory environment and fitted with motion-capture markers

to record their movements. These settings may have contributed to the relatively high attrition rate in this study. It is possible that the number of infants across clusters was affected by the attrition rate. Despite this, we were able to analyze a wide range of behavioural patterns due to the large sample size.

## Conclusion

We demonstrated large inter-individual differences in infants' behaviour when interacting with their environment. These differences reflect individually unique intrinsic dynamics for spontaneous movements, which are harnessed to produce instrumental behaviours. The present study indicates that elucidating the learning process in early infancy requires analysing not only the post-learning state but also the entire behavioural trajectory, including the pre-learning state.

## Data availability

All data supporting the findings of this study are provided at Supplementary Information and available at https://osf.io/n5h9u/?view_only=eff2ef083d024709ab79518c0f4c46d7.

## Code availability

The code for the analysis is publicly available at https://github.com/ryo-fujihira/Individual-differences-in-infants-behavioural-change.

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

## Acknowledgements

We thank Fumitaka Homae, Kayo Asakawa, and Yoshiko Koda for data collection; and Keiko Hirano, Tomoko Yoneyama, and Nobue Kanaya for administrative assistance. The part of this study was funded by Japan Society for Promotion of Science Grants-in-Aid for Scientific Research (24KJ0734 to R.F. and 23H05425 to G.T.). The funder had no role in study design, data collection and analysis, decision to publish or preparation of the manuscript.

## Author contributions

H.W. and G.T.: Conceived and designed the experiments. H.W. and G.T.: Performed the experiments. R.F., H.W. and G.T.: Analysed the data. R.F., H.W. and G.T.: Contributed materials/analysis tools. R.F. and G.T.: Wrote the paper.

## Competing interests

The authors declare no competing interests.
