## [Transparent Peer Review file · Communications Psychology]

Individual differences in how infants change behaviors from spontaneous to instrumental

Corresponding Author: Dr Ryo Fujihira

Version 0:

Decision Letter:

Dear Mr Fujihira,

Thank you for your patience during the peer-review process. I am sincerely sorry for the unconscionably long delay in returning to you with a decision. Your manuscript titled "Individual differences in how infants change behaviors from spontaneous to instrumental" has now been seen by 2 reviewers, and I include their comments at the end of this message. They find your work of interest but raised some important points. We are interested in the possibility of publishing your study in Communications Psychology, but would like to consider your responses to these concerns and assess a revised manuscript before we make a final decision on publication.

We therefore invite you to revise and resubmit your manuscript, along with a point-by-point response to the reviewers. Please highlight all changes in the manuscript text file.

Editorially, we consider it critical that you address methodological concerns through suitable control analyses. We also ask that the presentation is significantly improved to ensure the necessary clarity and to detail exactly how the present work confirms, contradicts, or extends existing literature.

I am attaching an Editorial Requests Table that details critical reporting requirements for the revised manuscript. Please attend to each item and ensure your manuscript is fully compliant. If your revised manuscript is not aligned with these requests on major issues, such as those concerning statistics, it may be returned to you for further revisions without re-review.

Please submit the following items:

- Revised manuscript
- Point-by-point response to the referees' comments
- Cover letter (as a separate document)
- <https://www.nature.com/documents/nr-reporting-summary.zip>>Nature Research Reporting Summary
- <https://www.nature.com/documents/nr-editorial-policy-checklist.pdf>>Editorial Policy Checklist
- Completed Editorial Request Table (attached).

via this link: Link Redacted .

Additional guidance is available in our style and formatting guide Communications Psychology formatting guide.

Best regards,

Marike Schiffer

Marike Schiffer, PhD
Chief Editor
Communications Psychology

REVIEWER EXPERTISE:

Reviewer #1 infant motor control

Reviewer #2 infant motor control

REVIEWER REPORTS:

Reviewer #1 (Remarks to the Author):

This manuscript examines the interindividual variations occurring in 2- and 3-months old infants while learning a mobile conjugate reinforcement task. One central point of the paper is to illustrate how group averages that often lead to monotonic learning curves, in fact are far from representing the actual individual learning curves. Several varying learning curves can be detected in subgroups of the population reflecting different rates of learning, and these variations cannot be solely attributed to age, but are more specific to other factors such as initial conditions, the type of exploratory behavior, and additional intrinsic factors that can determine the way infants interact with their environment and learn tasks. The method and analyses are clear, and the manuscript is well written.

However, although this reviewer really appreciates the work presented and the important points made, the overall message from this work is not exactly new, and its presentation also comes across as somewhat limited to the task context studied. Similar arguments have been made in prior studies and by researchers in other tasks and learning contexts with infants of other ages, especially from researchers familiar with the dynamic systems approach. The authors could make a much stronger argument about variations in developmental processes and learning by referring or linking their findings to these other studies. In the current manuscript version, it is hard to see how this argument extends beyond the ages and conjugate reinforcement task being used. Although it makes sense to focus on this task for the analyses given the data available, it misses generalizing the findings to the more fundamental variability property of development and learning that spans different tasks and ages. By highlighting other works making similar points, this paper would send a more powerful message about processes of development, a message that is often neglected when studies mainly focus on group averages.

Here are a few suggestions:

Jacobsohn et al. (2014) in *Developmental Psychobiology* have used clustering analyses to reveal distinct groups with distinct developmental trajectories that were missed by monotonic group averages. These analyses were applied to the development of hand preference from birth to 24 months.

Corbetta et al 2000, *Infant Behavior and Development* using a reaching task with 5- to 9-month-olds, highlighted variations in learning and response adaptation that were mainly pattern related, not age related.

Thelen et al., 1996, *JPE:HPP* on infant reaching, and Snapp-Childs and Corbetta 2009 on learning to walk illustrated how different learning strategies or learning curves end up coalescing and become indistinguishable from one another over time. Kurt Fisher has also written several theoretical papers on skill learning from a dynamic perspective applied to the field of education in infants and children that are quite inspiring. One that may be relevant is Fischer, K. W., & Rose, S. P. (1994). *Dynamic development of coordination of components in brain and behavior: A framework for theory and research*. In G. Dawson & K. W. Fischer (Eds.), *Human behavior and the developing brain* (pp. 3–66). New York: Guilford Press. But other of his papers also tackle the complexity of development and its change over time.

See also Stergiou, N., Yu, Y., & Kyvelidou, A. (2013). A perspective on human movement variability with applications in infancy motor development. *Kinesiology Review*, 2(1), 93-102, and other papers from Stergiou on this theme of movement variability.

Comments on terminology:

Why are the authors calling the phases when the arm is connected to the mobile as “play” phases, and not “acquisition” phases as typically referred to in the literature for this kind of task? The task is about learning the contingency between arm movements and mobile activation. The word play does not convey necessarily that infants are learning the contingency, while acquisition does. The authors themselves sometimes discuss their findings in terms of “acquiring instrumental actions” or “reinforcement”.

There should be a clear distinction in terminology between the spontaneous movements produced during baseline and the movements produced during the acquisition phase, but sometimes those phases appear confounded in the writing by using the term spontaneous movements to both phases. Maybe the authors could use the term “limb movement” instead of “spontaneous movements” when referring to movements produced during the acquisition phase. If movement frequency is increased to activate the mobile, maybe those are not simply spontaneous movements, but movements produced with the purpose of activating the mobile.

Finally, the authors use the term “individual differences” to refer to variability in development. In some subfields of psychology, individual difference is used to refer to long lasting inherent characteristics or traits of a person, such as their temperament, personality, etc. That is different from interindividual variability, which would be a preferred word usage in the context of this study. Indeed, individual differences would assume that the unique underlying behavioral traits observed on day 1 in one infant would remain constant over time, and observed again on day 2, 3, 4, etc., which cannot be demonstrated in this study. Interindividual variation on the other hand, points to the fact that infants differ in their learning rate without assuming any inherent or underlying personality or temperamental traits.

Other questions:

Why didn't the authors have an extinction phase at the end of the acquisition?

How do the researchers interpret the fact that their findings show increase in activity in the non-connected limbs as well during acquisition? How does that result speak to action/limb selection? See studies from Chinn, Somogy, Lockman on the use of buzzer to study body mapping. This group finds widespread activation across limbs in 3mo even when the buzzer is on one limb only.

The attrition rate seems high in this study. How does that compare to other studies using this same paradigm with young infants?

Thanks for an interesting paper.

Reviewer #2 (Remarks to the Author):

Overall this is a strong and interesting paper on early learning and movement. The flow from group data to individual data to simulation data are strong. My review asks a lot of questions to the authors as some areas need clarity. Most notably are the 1) absence of extinction data in an operant conditioning paradigm; 2) unclear use of individual data from prior literature (baseline ratio of 1.5 to indicate learning); and 3) unclear use of displacement as a measure of contingency when small displacements could also move the mobile.

Abstract

Does instrumental mean the same thing as voluntary or task-specific?

In some classic mobile paradigm literature individual learning metrics are used. For example, 1.5x the kicking rate in acquisition or extinction is an individual baby threshold. This needs to be rectified with the statement on group analysis (which are more common).

Define monotonic increases

Intro

Recent work shows that spontaneous movement may inform the sensory system and then affect the motor cortex. How does this work fit in that context.

With the cluster analysis and dynamic systems approach it seems like the focus on spontaneous (vs. purposeful) movements was not an a priori hypothesis. But the introduction seems like it was.

Results

For the conventional analysis. Some sentences of general results and interpretation are needed. Did all limbs increase over time with preferential increase in the tethered limb? And then some specific baseline-to-phase increases in the post hoc analysis? And the 3 month olds out performing the 2 month olds?

Extinction is ignored. Why is there no Extinction phase?

Why isn't the baseline ratio for baseline = 1

For sentence like: In addition, significant interaction between the months and phases was observed ($F(5, 915) = 4.70, p < .001$). Please change “months” to “age” or say “months of age”. Also, sometimes, the P for Phase and the P for Play are confusing.

The within groups analysis for main effects and post hoc phase comparisons does not add more than the 2 age groups analysis and could be cut.

Better justify the rationale for not combining the 2- and 3-month-olds in the overall analysis (or at least after the conventional analysis). It is understood this is done for the cluster analysis.

Displacement: Clarify why a larger displacement is expected. A larger baseline ratio is understood. Displacement needs more explanation. Wouldn't a lot of smaller displacements also move the mobile? If so, did infants not choose this strategy over time?

In many of the mobile paradigm papers (traditional, leg as the tethered limb) – they somewhat ignore the earlier acquisition

phases and focus on later phases where there is a more pronounced behavioral change. In addition, there is somewhat of an allowance because learning the contingency takes more than a few repetitions (or minutes). Given this information, why does this paper include the early play phases?

The cluster analysis for cluster 4 with N=9 is interesting. But, these participants demonstrate kicking well above the 1.5 baseline ratio that is observed in other studies. And there is no Extinction.

A decrease in movement may also be a way of learning a contingency. Or an exploratory behavior.

I can't see the supplementary figures and have a feeling they need to go in the main body of the manuscript.

Version 1:

Decision Letter:

Dear Dr Fujihira,

Your manuscript titled "Individual differences in how infants change behaviors from spontaneous to instrumental" has now been seen by our reviewers, whose comments appear below. In light of their advice I am delighted to say that we are happy, in principle, to publish a suitably revised version in Communications Psychology.

We therefore invite you to revise your paper one last time to address the remaining concerns of our reviewers and a list of editorial requests. At the same time we ask that you edit your manuscript to comply with our format requirements and to maximise the accessibility and therefore the impact of your work.

EDITORIAL REQUESTS:

SUBMISSION INFORMATION:

OPEN ACCESS:

* **DATA AVAILABILITY:**

Link Redacted

Best regards,

Marike

Marike Schiffer, PhD
Chief Editor
Communications Psychology

REVIEWERS' COMMENTS:

Reviewer #1 (Remarks to the Author):

I appreciate the authors' revisions to this manuscript and clarifications provided to my comments. This is a great study providing a wonderful approach of much needed analyses that are often neglected in developmental psychology. Thank you for providing such clear insights into infant motor development. I recommend acceptance.

Reviewer #2 (Remarks to the Author):

Major Comment

1. Make sure the intro makes it clear that the vast majority of the history of the MP and literature used in this paper are with MP where the infant's legs are tethered to the mobile and this study uses the arms. The discussion makes this clearer than the intro.

Minor Comments

My comments are mostly for clarity since this paper introduces newer terminology for the mobile paradigm (play vs. acquisition) and uses cluster analysis and a dynamics systems approach.

1. **Reword sentence for clarity:**

While the group-averaged data showed a gradual increase in arm movements, individual learning curves exhibited few simple monotonic gradual increases and were much more complex.

2. The phrase "intrinsic movement repertoires" is better but I do worry that intrinsic might be interpreted as obligatory. How about initial movement repertoires (which is a combination of the original term "initial conditions" and the new one.

3. Take out the word monotonous in this sentence if it is unnecessary (I see the link in the discussion, but still)
However, a more detailed analysis of pre-learning states and changes during the learning process have suggested that learning involves dynamic properties that cannot be solely captured by the monotonic reinforcement of movement.
4. Refer to the mobile paradigm consistently vs. mobile task and others iterations
Example: conduct a mobile paradigm task experiment, analyzing individual differences in the time evolution of their movements
5. Reword for clarity:
When infants were As the infants were likely to alert and playful, the data collection started. measurements were carried out.
6. I like the new section that includes this sentence on page 8-9. But, something is missing from this sentence as it does not make sense to me.
“the full trajectory of behaviour in infants of different ages—from baseline, through the moment the mobile was connected, and including sufficient time for the infant to learn the contingency with the environment.”
7. Clarify if this sentence “In other words, the individual trajectories of behavioural change evolved from a state unique to each cluster to one indistinguishable from others.” It is reading like the mobile paradigm, as an identical task all infants were exposed to, produced similar behaviors (and therefore possibility less variability than baseline where there were no task constraints).
8. Overall: very strong and interesting paper

In cases where reviewers are anonymous, credit should be given to 'Anonymous Referee' and the source.
The images or other third party material in this Peer Review File are included in the article's Creative Commons license, unless indicated otherwise in a credit line to the material. If material is not included in the article's Creative Commons license and your intended use is not permitted by statutory regulation or exceeds the permitted use, you will need to obtain permission directly from the copyright holder.

Reviewer #1 (Remarks to the Author):

We thank Reviewer #1 for the thoughtful comment. The reviewer's comments have been helpful to clarify the positioning of our paper. The revisions made in response to the reviewer's comments have improved the paper's contribution to a broader understanding of learning phenomena, beyond the mobile paradigm. We also greatly appreciate the various references you kindly provided.

This manuscript examines the interindividual variations occurring in 2- and 3-months old infants while learning a mobile conjugate reinforcement task. One central point of the paper is to illustrate how group averages that often lead to monotonic learning curves, in fact are far from representing the actual individual learning curves. Several varying learning curves can be detected in subgroups of the population reflecting different rates of learning, and these variations cannot be solely attributed to age, but are more specific to other factors such as initial conditions, the type of exploratory behavior, and additional intrinsic factors that can determine the way infants interact with their environment and learn tasks. The method and analyses are clear, and the manuscript is well written.

However, although this reviewer really appreciates the work presented and the important points made, the overall message from this work is not exactly new, and its presentation also comes across as somewhat limited to the task context studied. Similar arguments have been made in prior studies and by researchers in other tasks and learning contexts with infants of other ages, especially from researchers familiar with the dynamic systems approach. The authors could make a much stronger argument about variations in developmental processes and learning by referring or linking their findings to these other studies. In the current manuscript version, it is hard to see how this argument extends beyond the ages and conjugate reinforcement task being used. Although it makes sense to focus on this task for the analyses given the data available, it misses generalizing the findings to the more fundamental variability property of development and learning that spans different tasks and ages. By highlighting other works making similar points, this paper would send a more powerful message about processes of development, a message that is often neglected when studies mainly focus on group averages. Here are a few suggestions:

Jacobsohn et al. (2014) in *Developmental Psychobiology* have used clustering analyses to reveal distinct groups with distinct developmental trajectories that were missed by monotonic group averages. These analyses were applied to the development of hand preference from birth to 24 months.

Corbetta et al 2000, *Infant Behavior and Development* using a reaching task with 5- to 9-month-

olds, highlighted variations in learning and response adaptation that were mainly pattern related, not age related.

Thelen et al., 1996, JPE:HPP on infant reaching, and Snapp-Childs and Corbetta 2009 on learning to walk illustrated how different learning strategies or learning curves end up coalescing and become indistinguishable from one another over time.

Kurt Fisher has also written several theoretical papers on skill learning from a dynamic perspective applied to the field of education in infants and children that are quite inspiring. One that may be relevant is Fischer, K. W., & Rose, S. P. (1994). Dynamic development of coordination of components in brain and behavior: A framework for theory and research. In G. Dawson & K. W. Fischer (Eds.), *Human behavior and the developing brain* (pp. 3–66). New York: Guilford Press. But other of his papers also tackle the complexity of development and its change over time.

See also Stergiou, N., Yu, Y., & Kyvelidou, A. (2013). A perspective on human movement variability with applications in infancy motor development. *Kinesiology Review*, 2(1), 93-102, and other papers from Stergiou on this theme of movement variability.

Thank you for your warmful and constructive feedback. I fully agree that incorporating a more general description based on the suggested literature is important. We have incorporated additional introduction and discussions, in relation to our study. In the introduction, we incorporated all the references you suggested and clarified the relationship between our study and the previous studies on individual differences as follows:

p.3. 'Previous studies have demonstrated individual differences in the developmental trajectories of motor and behavioural skills over timescales of days, months, or years. These include the acquisition of reaching (Jacobson et al., 2014; Corbetta et al., 2000; Thelen et al., 1996), the development of crawling (Kobayashi et al., 2021), sitting (Stergiou et al., 2013), and walking (Snapp-Childs & Corbetta, 2009; Stergiou et al., 2013), as well as more general processes of skill learning (Rose & Fischer, 1998). However, inter-individual differences in learning over shorter timescales have not been clearly identified. Short-term learning is not a mere fluctuation for development, but potentially drive longer-term development through memory processes (Fujihira and Taga 2025), and dynamic interactions between brain, body, and environment.'

We could not get the Kurt Fischer's book, instead we red and referred the following article. Rose, S. P., & Fischer, K. W. (1998). Models and rulers in dynamical development. *British Journal of Developmental Psychology*, 16(1), 123-131.

Also, we added the following general discussions.

p.27. 'Clustering analyses with different tasks and different age groups have also revealed various learning trajectories at the individual level hidden under the averaged description, such as hand preference from birth to 24 months (Jacobsohn et al. 2014), and maturation of walking skills during a several months from the onset of independent upright locomotion (Snapp-Childs and Corbetta 2009). Consistent with our findings, subgroup analyses in these studies elucidated that initially diverse states gradually converged into similar patterns with reduced individual difference through interactions with the environment. This convergence is considered a nomothetic phenomenon in learning processes, and the present study further demonstrated that such a shift can emerge even within a brief timescale of approximately ten minutes.'

p.28. 'The integration of these mechanisms constitutes the dynamics of individual motor behaviour, and serve as a determinant of both the selection of behavioural strategies (Corbetta et al. 2000) and the degree of movement maturity (Thelen et al. 1996) across development.'

Also, related to the other reviewer's comment, I added another general discussion as follows:

p.28. 'A recent study demonstrated that spontaneous movements generate sensorimotor coordination even in the absence of specific goals toward the environment (Kanazawa et al. 2023), contributing to the formation of unique movement attractors in each infant. Given that the spontaneous formation of these attractors occurs without a predefined direction, it is likely to exhibit considerable diversity. However, once interaction with the environment begins, the movement dynamics are modified to adapt to environmental contingencies, potentially leading to a reduction of individual differences. The emergence of rhythmic and stereotypical movements during this developmental period (Thelen 1979, 1981a, 1981b) may provide opportunities for sensorimotor adaptation by promoting interaction with the environment through the use of available motor repertoires of spontaneous movements (Watanabe and Taga 2006).'

Comments on terminology:

Why are the authors calling the phases when the arm is connected to the mobile as "play" phases, and not "acquisition" phases as typically referred to in the literature for this kind of task? The task is about learning the contingency between arm movements and mobile activation. The word play does not convey necessarily that infants are learning the contingency, while acquisition does. The authors themselves sometimes discuss their findings in terms of "acquiring instrumental actions" or "reinforcement".

Thank you for the important comment. We used 'play' instead of 'acquisition' to express that learning in this paradigm is not necessarily restricted to acquiring specific movements. I added such explanation in the main text as follows:

p.8-p.9. 'In this context, we named the phase typically referred to as acquisition phase as play phase. This terminology reflects a broader conceptualization of learning—not only the acquisition of specific movements but also including a variety of intermediate behavioural changes.'

There should be a clear distinction in terminology between the spontaneous movements produced during baseline and the movements produced during the acquisition phase, but sometimes those phases appear confounded in the writing by using the term spontaneous movements to both phases. Maybe the authors could use the term "limb movement" instead of "spontaneous movements" when referring to movements produced during the acquisition phase. If movement frequency is increased to activate the mobile, maybe those are not simply spontaneous movements, but movements produced with the purpose of activating the mobile.

Thank you for the important comment. Spontaneous movements refer to actions generated by endogenous neural activity that are not directly linked to specific sensory inputs or goals. However, it is difficult to completely eliminate the influence of external factors, such as sensory input from the environment. Moreover, even goal-directed movements are thought to be generated based on the same neural mechanisms that underlie spontaneous movements. Therefore, to avoid the confusion pointed out by the reviewer, in the discussion section (p.23 & p.24), I changed 'spontaneous movements' to 'limb movements', which refer to movements produced during the play phase.

Finally, the authors use the term "individual differences" to refer to variability in development. In some subfields of psychology, individual difference is used to refer to long lasting inherent characteristics or traits of a person, such as their temperament, personality, etc. That is different from interindividual variability, which would be a preferred word usage in the context of this study. Indeed, individual differences would assume that the unique underlying behavioral traits observed on day 1 in one infant would remain constant over time, and observed again on day 2, 3, 4, etc., which cannot be demonstrated in this study. Interindividual variation on the other hand, points to the fact that infants differ in their learning rate without assuming any inherent or underlying personality or temperamental traits.

Thank you for the important comment. In response to your initial comment, we have added an explanation about individual differences in the introduction. Please refer to that section.

In addition, from a dynamical perspective, infants have their own attractors in the spontaneous movement. If the attractors have a same shape across individuals, the difference in movements is merely the result of variation. However, it is known that spontaneous movements differ across infants and exhibit consistent characteristics within individuals, as written in the manuscript:

p.3. 'In particular, the behaviour of young infants is characterized by varieties of spontaneous movements, which evolve rapidly over the postnatal period¹⁶ ~~and~~ but exhibit intra-individual consistency alongside significant inter-individual differences¹⁷⁻¹⁹.'

Also we added sentences explaining 'individual differences' in spontaneous movements from the dynamical systems perspective as follows:

p.6. 'Observed behavioural states are generated from a dynamical system composed of infant's brain, body, and environment, containing the overall characteristics of each infant's movement dynamics. Therefore, the behavioural trajectory of each individual is considered to reflect the individual differences in the structurally stable attractor dynamics underlying movement generation.'

Also, we added the following reference.

19. Groome, L. J., Swiber, M. J., Holland, S. B., Bentz, L. S., Atterbury, J. L., & Trimm III, R. F. (1999). Spontaneous motor activity in the perinatal infant before and after birth: Stability in individual differences. *Developmental Psychobiology: The Journal of the International Society for Developmental Psychobiology*, 35(1), 15-24.

Thus, each infant has his/her own attractor dynamics in the spontaneous movement and a single measurement of these movements can reflect the underlying differences in the attractor dynamics which referred to as 'individual difference' in this manuscript.

However, we do not refer to temperament or personality in this article, thus, we increased the use of expressions such as 'individual difference in behavioural changes' throughout the manuscript.

Other questions:

Why didn't the authors have an extinction phase at the end of the acquisition?
--

Thank you for the comment. It is because our primary objective was to investigate the variety of behavioural changes from spontaneous to instrumental. We clarified this reason in the method section as follows:

p.8. 'In the mobile paradigm, an extinction phase is typically introduced at the end by removing the connection between the infant's limb and the mobile. In the present study, however, we extended the play phase instead, as our primary objective was to investigate the variety of behavioural trajectories from spontaneous to instrumental.'

How do the researchers interpret the fact that their findings show increase in activity in the non-connected limbs as well during acquisition?

Thank you for the question. I added interpretation about non-connected limbs increase as follows related to reproducibility challenges (underlined):

p.26-p.27. 'Our clustering analysis showed that two-third of 3-month-old infants did not differentiate their arm and leg movements (cluster 1 and 3), whereas remaining one-third of 3-month-old infants (cluster 2 and 4) differentiated them. The inter-cluster differences are related to whether there is a bias in the amount of spontaneous movement between the arms and legs. Taking into account the individual differences in spontaneous movement patterns, it is suggested that the process of learning to move the mobile involves both age-related differences and inter-individual differences, and the behavioural changes related to learning are not necessarily limited to the connected limb alone.'

How does that result speak to action/limb selection? See studies from Chinn, Somogy, Lockman on the use of buzzer to study body mapping. This group finds widespread activation across limbs in 3mo even when the buzzer is on one limb only.

Thank you for the comment. I already referred their study as Somogyi et al. 2018 and discussed that some studies reported that limb movement differentiation is observed in 3 or 4 months of age^{23,27,45}, while other studies showed no such evidence^{3,56} (p.26). Then, I suggest that experiments involving a sufficiently large sample of infants within a restricted age range and subgroup analysis like our clustering can solve this reproducibility challenge.

In the manuscript, I added the following sentences.

p.26-p.27. 'The inter-cluster differences are related to whether there is a bias in the amount of spontaneous movement between the arms and legs.'

~~'This suggests that~~ Recruiting a sufficient number of infants within a restricted age range and extracting subgroups based on the individuality of infants' spontaneous movements could lead to more reproducible analytical results.'

The attrition rate seems high in this study. How does that compare to other studies using this same paradigm with young infants?

The unfamiliar laboratory setting including motion capture system, along with the prolonged play phase, may have influenced the attrition rate. Nonetheless, the selection procedure was clearly defined and strictly followed. Given that the experimental setting and procedures are unique to our study, a straightforward comparison of attrition rates with other studies employing similar paradigms is difficult. However, we added the issue of attrition rate in the limitations as follows:

p.29. 'Additionally, in our measurement, the infants were placed in an unfamiliar laboratory environment and fitted with motion-capture markers to record their movements. These settings may have contributed to the relatively high attrition rate in this study. It is possible that the number of infants across clusters was affected by the attrition rate. Despite this, we were able to analyze a wide range of behavioral patterns due to the large sample size.'

Reviewer #2 (Remarks to the Author):

We thank Reviewer #2 for insightful comments. In particular, the reviewer's comments have been helpful to refine the logic of our paper. We have also been able to replace incorrect terminology. Thus, the revision made in response to the reviewers' comments have improved the paper's contribution to a more precise understanding of phenomena and readability for the audience.

Overall this is a strong and interesting paper on early learning and movement. The flow from group data to individual data to simulation data are strong. My review asks a lot of questions to the authors as some areas need clarity. Most notably are the 1) absence of extinction data in an operant conditioning paradigm; 2) unclear use of individual data from prior literature (baseline ratio of 1.5 to indicate learning); and 3) unclear use of displacement as a measure of contingency when small displacements could also move the mobile.

Thank you for the kind words. All your questions have improved our paper's contribution. Below are the responses to the three main questions. Please note that there may be some overlap within the answers for following separated questions.

- 1) It is because our primary objective was to investigate the variety of behavioural changes from spontaneous to instrumental. We clarified it in the method section as follows:
p.8. 'In the mobile paradigm, an extinction phase is typically introduced at the end by removing the connection between the infant's limb and the mobile. In the present study, however, we extended the play phase instead, as our primary objective was to investigate the variety of behavioural trajectories from spontaneous to instrumental.'

We also added a paragraph in Discussion to describe the issue in relation to the extinction phase as follows:

p.24-p.25. 'In the context of mobile paradigm, an extinction phase has been used as a procedure to examine whether behaviour is generated based on memory by measuring baseline ratio or retention ratio (Rovee-Collier and Fagen 1976, Rovee-Collier et al. 1980). However, this procedure has its limitations. One major weakness is that learning in a certain number of infants cannot be assessed using the ratio in the amount of movement. Additionally, some infants may quickly realize that moving their arms no longer causes the mobile to move, and consequently stop the behaviour they had previously learned and generate another behavioural change. In other words, it cannot be ruled out that a new

learning process may begin under these altered environmental conditions. In the present study, our focus was to elucidate the diverse behaviours exhibited by infants to learn the contingency between their own movements and the environmental changes during a sufficiently long play phase. Therefore, we did not include an extinction phase. While classical studies on memory using the mobile paradigm have provided foundational insights (Rovee-Collier and Fagen 1976, Rovee-Collier et al. 1980), a recent theoretical development has begun to shed new light on the dynamics of memory (Fujihira & Taga 2025)' and future research should include finer-grained analyses of movement changes on shorter timescales⁴ and additional internal state measures such as EEG^{45,46}.

2) I added an explanation for the learning threshold as follows:

p.4. 'Based on these results, it has become customary to use an increase of 1.5 times the baseline level of limb movement as the threshold for learning.'

In addition, to clarify our position that learning is not limited to an increase in movement, we have also added the following sentence to the Introduction.

p.4-p.5. 'These studies imply that learning is not solely reflected in increased limb movement.'

3) I think you're absolutely right. We are not necessarily expecting large displacements. Our approach is to treat all behavioural changes occurred during the task as learning process, and we used the displacement rate as a means to analyze those changes.

we have added explanations of displacement rates in the method and result section as follows:

p.9. 'The displacement rate can be considered a general measure of movement quantity, as it captures changes in both the magnitude and frequency of movements, and it has been shown to correlate with the number of movement units (Watanabe and Taga 2006).'

p.13-p.14. 'Displacement rates were calculated to measure the amount of movement in each limb within a one-minute time window. This measure reflects the extent of limb movement during a given time window while its ratio to the baseline level (baseline ratio) indicates the degree of increase or decrease in limb movement relative to the baseline phase.'

Abstract

Does instrumental mean the same thing as voluntary or task-specific?
--

Thank you for your important question. We think the meaning of instrumental is most closely aligned with that of goal-directed, and it is similar to voluntary or task-specific. We added sentences about the meaning of instrumental and why we used it as follows in introduction:

p.4. 'As a result of learning contingency between their own movements and mobile movements, infants change their spontaneous movements into actions suitable for moving the mobile. We refer to the learned actions as instrumental behaviours (Thelen and Fisher 1983), as they are performed to achieve a specific outcome. Although such behaviours are probably goal-directed⁴, it is uncertain whether infants have a goal when performing these actions. Thus, we adopt the term "instrumental."

In some classic mobile paradigm literature individual learning metrics are used. For example, 1.5x the kicking rate in acquisition or extinction is an individual baby threshold. This needs to be rectified with the statement on group analysis (which are more common).

Thank you for the insightful suggestion. Explaining the background behind the use of learning criteria is important; however, since this is specific to the mobile paradigm, we have added this point to the Introduction as follows:

p.4. 'Based on these results, it has become customary to use an increase of 1.5 times the baseline level of limb movement as the threshold for learning.'

In addition, to clarify our position that learning is not limited to an increase in movement, we have also added the following sentence to the Introduction.

p.4. 'These studies imply that learning is not solely reflected in increased limb movement.'

Define monotonic increases

Thank you for the advice about readability of the abstract. I changed the word 'monotonic' in this abstract and some in the main text to 'gradual'. This word is simple but can describe similar meaning as 'monotonic'.

Intro

Recent work shows that spontaneous movement may inform the sensory system and then affect the motor cortex. How does this work fit in that context.

Thank you for the suggestion. In line with your comment, spontaneous movements are considered to contribute to sensorimotor adaptation. I added related discussion as follows:
p.28. 'A recent study demonstrated that spontaneous movements generate sensorimotor coordination even in the absence of specific goals toward the environment (Kanazawa et al. 2023), contributing to the formation of unique movement attractors in each infant. Given that the spontaneous formation of these attractors occurs without a predefined direction, it is likely to exhibit considerable diversity. However, once interaction with the environment begins, the movement dynamics are modified to adapt to environmental contingencies, potentially leading to a reduction of individual differences. The emergence of rhythmic and stereotypical movements during this developmental period (Thelen 1979, 1981a, 1981b) may provide opportunities for sensorimotor adaptation by promoting interaction with the environment through the use of available motor repertoires of spontaneous movements (Watanabe and Taga 2006).'

With the cluster analysis and dynamic systems approach it seems like the focus on spontaneous (vs. purposeful) movements was not an a priori hypothesis. But the introduction seems like it was.

Thank you very much for the insightful comment. Actually, we had such a hypothesis in advance, but it was misleading to describe spontaneous movement as 'initial condition' for the subsequent learning. We revised the introduction to convey that spontaneous movements are important for learning, in the sense that both spontaneous and learned movements are generated from the same movement repertoires as follows:

p.3-4. 'Through these movements, infants can initiate interactions with the physical world leading to acquire actions along their unique ways^{20,21}. Thus, different patterns of spontaneous movements can provide different foundations for shaping actions toward environments. ~~servng as dynamic and diverse initial conditions for behavioural changes.~~ In the context of learning, we cannot ignore the fact that ~~initial conditions~~ intrinsic movement repertoires may differ from infant to infant, which raises the following question: how do the patterns of intrinsic spontaneous movements affect learning.

To examine the dynamic evolution of behaviours during learning, ~~The most appropriate experimental setting for addressing this question is the~~ we leveraged an experimental paradigm known as the mobile paradigm, which has widely been used for revealing learning capabilities in early infancy ^{1,4,22-25}.'

Results

For the conventional analysis. Some sentences of general results and interpretation are needed. Did all limbs increase over time with preferential increase in the tethered limb? And then some specific baseline-to-phase increases in the post hoc analysis? And the 3 month olds out performing the 2 month olds?

Thank you for the suggestion to improve our results section. We described the interpretation of the results in the following paragraph after presenting the statistical analysis. However, we now realize that having a paragraph break is misleading. Thus, we removed the paragraph break after the sentence on p.14 'Changes were also observed in the limbs not connected to the mobile.', and it will make the flow more coherent.

Extinction is ignored. Why is there no Extinction phase?

This has already been addressed in our response to the main point raised.

Why isn't the baseline ratio for baseline = 1

This is because the baseline ratio is calculated using displacement rate calculated in a one-minute time window. The average of baseline ratio over the 2-minute baseline period is set to 1, but it does not necessarily equal to 1 in a one-minute time window.

We added the following sentence in the caption of Fig. 1.

'Thus, the average of baseline ratio over the baseline phase is equal to 1.'

For sentence like: In addition, significant interaction between the months and phases was observed ($F(5, 915) = 4.70, p < .001$). Please change "months" to "age" or say "months of age". Also, sometimes, the P for Phase and the P for Play are confusing.

Thank you for the kind advice. I changed "months" to "age", and "Phase" written in the Fig. 2 and Fig. 6 to "2-min blocks".

The within groups analysis for main effects and post hoc phase comparisons does not add more than the 2 age groups analysis and could be cut.

It is true that the text contains a large amount of statistical information and may appear verbose, but we believe this level of detail is necessary to demonstrate the overall increase in movement for each age group.

Better justify the rationale for not combining the 2- and 3-month-olds in the overall analysis (or at least after the conventional analysis). It is understood this is done for the cluster analysis.

Thank you for the advice.

We added the following sentence in the last paragraph of the introduction and the first result section to justify the rationale for combining the age groups in the subsequent analysis.

p.5. 'This analysis further investigates whether age differences alone can fully account for individual differences in learning.'

p.16. 'Furthermore, there are a large magnitude of standard deviations in all panels, which indicates that ~~there are~~ age differences alone cannot reveal the covert large inter-individual differences in behavioural changes.'

Displacement: Clarify why a larger displacement is expected. A larger baseline ratio is understood. Displacement needs more explanation. Wouldn't a lot of smaller displacements also move the mobile? If so, did infants not choose this strategy over time?

This has already been addressed in our response to the main point that you raised.

In many of the mobile paradigm papers (traditional, leg as the tethered limb) – they somewhat ignore the earlier acquisition phases and focus on later phases where there is a more pronounced behavioral change. In addition, there is somewhat of an allowance because learning the contingency takes more than a few repetitions (or minutes). Given this information, why does this paper include the early play phases?

In previous studies using the mobile paradigm, relatively long play phases were often employed when the experiments were conducted at home. In contrast, when conducted in laboratory settings including those of our previous studies, the play phase typically lasted around six minutes. Nevertheless, behavioral changes within that duration have been consistently reported. For infants, being placed alone in a novel laboratory environment away from their caregivers can make even a few minutes period appear quite lengthy. In the present study, the 10 minutes following the two-minute baseline phase must be experienced as challenging for some of the infants. Therefore, this study implemented observation over the long period,

capturing the full trajectory of behavior in infants of different ages—from baseline, through the moment the mobile was connected, and including sufficient time for the infant to learn the contingency with the environment. As a result of the clustering analysis, we found that the behavioural changes in the early play phase became apparent.

To make this point clear, I added a sentence at the beginning of results for clustering as follows:

p.17. 'To uncover individual differences in the learning process, it is essential to analyse the entire dynamics of behavioural trajectories, including early-phase changes.'

I also added a sentence highlighting the importance of clustering in relation to the behavioural changes in early phase as follows:

p.20. 'The inter-cluster differences in the baseline phase disappeared at the P1 phase by the increase (cluster 2 and 4) and the decrease (cluster 1) in the amount of limb movement.

Behavioural changes in the early play phase became apparent through this clustering analysis.'

I further edited the procedure (method section) as follows:

p.8. in order to capture 'the full trajectory of behavior in infants of different ages—from baseline, through the moment the mobile was connected, and including sufficient time for the infant to learn the contingency with the environment.'

The cluster analysis for cluster 4 with N=9 is interesting. But, these participants demonstrate kicking well above the 1.5 baseline ratio that is observed in other studies. And there is no Extinction.

Thank you for the comment. I added a sentence highlighting the behaviours of infants in cluster 4 as follows:

p.21. 'Additionally, individual data from infants in Cluster 4 showed a high baseline ratio (Fig. S5), while the actual amount of movement, measured by displacement rate (Fig. S6), was not much different from other clusters during the play phase. This also indicates that they exhibited a lower amount of spontaneous movement in the individual level.'

A decrease in movement may also be a way of learning a contingency. Or an exploratory behavior.

I can't see the supplementary figures and have a feeling they need to go in the main body of the manuscript.

I agree with your intuition. Observing individual behavioral trajectories is crucial for identifying such 'nonstandard' learning patterns. The supplementary material was indeed submitted, so it

might be a system-related issue. I would be very happy to show individual data with great complexity. However, the figure included in the supplement is quite large, so we think it would be difficult to include it in the main text.

Relatedly, we added limitations of our quantitative measurement and future directions of qualitative analysis as follows:

p.28-p.29. 'In the present study, we focused on the individual differences in the quantity of limb movements. However, a qualitative analysis of movement trajectories may further elucidate individual differences in infants' behavioural changes. Dimensionality reduction techniques such as PCA, as well as more recently developed methods like UMAP⁶² and t-SNE⁶³, may be useful for exploring the dynamic evolution of qualitative patterns. In addition, limb synergies have been reported as effective for analysing the patterns of limb movements⁴¹. Future studies should investigate individual differences in the qualitative patterns of infants' behavioural changes from spontaneous to instrumental using these techniques, leading to provide further support for the idea that the patterns of spontaneous movements are closely linked to learning.'

Reviewer #1 (Remarks to the Author):

I appreciate the authors' revisions to this manuscript and clarifications provided to my comments.

This is a great study providing a wonderful approach of much needed analyses that are often neglected in developmental psychology. Thank you for providing such clear insights into infant motor development. I recommend acceptance.

I appreciate your thoughtful comment. Your review has significantly improved the quality and value of my manuscript.

Reviewer #2 (Remarks to the Author):

Major Comment

1. Make sure the intro makes it clear that the vast majority of the history of the MP and literature used in this paper are with MP where the infant's legs are tethered to the mobile and this study uses the arms. The discussion makes this clearer than the intro.

I added the following sentence in Intro (p.5)

'Unlike previous studies that measured the number of kicks, this study attached a string to the arm and employed a motion analysis system to capture richer movement dynamics.'

Minor Comments

My comments are mostly for clarity since this paper introduces newer terminology for the mobile paradigm (play vs. acquisition) and uses cluster analysis and a dynamics systems approach.

1. Reword sentence for clarity:

While the group-averaged data showed a gradual increase in arm movements, individual learning curves exhibited few simple gradual increases and were much more complex.

I reworded this sentence as follows:

While the group-averaged data showed a gradual increase in arm movements, individual learning curves rarely exhibited ~~few~~ such simple gradual increases and instead displayed ~~were much~~ more complex patterns.

2. The phrase "intrinsic movement repertoires" is better but I do worry that intrinsic might be interpreted as obligatory. How about initial movement repertoires (which is a combination of the original term "initial conditions" and the new one).

I understand your perspective. However, from a dynamical systems perspective, considering that each infant possesses a unique attractor from which spontaneous movements emerge, the term "intrinsic" seems more appropriate than initial.

3. Take out the word monotonic in this sentence if it is unnecessary (I see the link in the discussion, but still)

However, a more detailed analysis of pre-learning states and changes during the learning process have suggested that learning involves dynamic properties that cannot be solely captured by the monotonic reinforcement of movement.

I removed the word monotonic from this sentence.

4. Refer to the mobile paradigm consistently vs. mobile task and others iterations
Example: conduct a mobile paradigm task experiment, analyzing individual differences in the time evolution of their movements

Thank you for the suggestion. In the revised manuscript, I consistently used “mobile paradigm” to generally denote the research method, and “mobile task” to describe the task actually performed by the infants in this study. As the expression “task experiment” is redundant, I replaced it with “mobile task”.

5. Reword for clarity:
When infants were As the infants were likely to alert and playful, the data collection started. measurements were carried out.

I reworded this sentence following your suggestion.

6. I like the new section that includes this sentence on page 8-9. But, something is missing from this sentence as it does not make sense to me.
“the full trajectory of behaviour in infants of different ages—from baseline, through the moment the mobile was connected, and including sufficient time for the infant to learn the contingency with the environment.”

I simplified this sentence to improve the readability as follows:

The measurement duration of the current study is longer than our past studies^{23,28,31,32,40,41} in order to capture the full trajectory of behaviour ~~in infants of different ages—from baseline, through the moment the mobile was connected, and including sufficient time for the~~ while infants ~~to~~ learn the contingency with the environment.

7. Clarify if this sentence “In other words, the individual trajectories of behavioural change

evolved from a state unique to each cluster to one indistinguishable from others.” It is reading like the mobile paradigm, as an identical task all infants were exposed to, produced similar behaviors (and therefore possibility less variability than baseline where there were no task constraints).

I added an explanatory sentence as follows (underlined):

‘In other words, the individual trajectories of behavioural change evolved from a state unique to each cluster to one indistinguishable from others. Environmental interactions serve to produce similar patterns with reduced individual differences.’

8. Overall: very strong and interesting paper

Thank you very much. I appreciate your thoughtful comments. your review has made the paper much more readable and will help it reach a wider audience.